# Edge displacement scores

Arne Melsom[1]

[1]Norwegian Meteorological Institute

**Correspondence:** Arne Melsom (arne.melsom@met.no)

**Abstract.** As a consequence of a diminishing sea ice cover in the Arctic, activity is on the rise. The position of the sea ice edge, which is generally taken to define the extent of the ice cover, changes in response to dynamic and thermodynamic processes. Forecasts for sea ice expansion on synoptic time scales due to an advancing ice edge will provide information that can be of significance for open ocean operations in polar regions. However, the value of this information depends on the quality of the forecasts. Here, we present methods for examining the quality of forecasted sea ice expansion on sub-seasonal time scales and the geographic location where the largest expansion are expected from the forecast results. The algorithm is simple to implement, and an examination of two years of model results and accompanying observations demonstrates the usefulness of the analysis.

## 1 Introduction

Due to climate change the sea ice extent is in decline in the Arctic (Parkinson, 2014). This change has led to increased activity in the region, and commercial shipping in open waters via Arctic sea routes will become increasingly economically viable in the 21$^{\text{st}}$ century (Aksenov et al., 2017). Thus, data sets for monitoring and forecasting sea ice conditions are receiving growing attention.

The past years have seen a flurry of activity related to assessing the quality of sea ice data sets. Dukhovskoy et al. (2015) presented a review and comparison of various traditional metrics for assessments of the skill of sea ice models. Goessling et al. (2016) introduced the Integrated Ice-Edge Error (IIEE), a quantity for describing mismatching sea ice extents from two data sets to analyze the predictability of the sea ice edge. Melsom et al. (2019) took advantage of the IIEE in their examination of various metrics for assessment of the quality of forecasts for the sea ice edge position. Methods for examining the quality of probabilistic results for sea ice conditions have been introduced by Goessling and Jung (2018) and Palerme et al. (2019). Recently, Cheng et al. (2020) examined the accuracy of a visually estimated ice concentrations monitoring product.

The changing position of the sea ice edge is generally not only shifted by dynamic advection, but can be significantly affected by the thermodynamics as well (Bitz et al., 2005). Thus, the temporal displacement of the sea ice edge will be affected by freezing along the perimeter of the sea ice extent in winter, and melting in summer. Hence, pattern-recognition algorithms

for displacements using maximum cross-correlation (MCC) methods such as those introduced by Leese et al. (1971) for wind vectors, and later for ocean surface currents (Tokmakian et al., 1990) and sea ice vectors (Lavergne et al., 2010), are not ideal for tracking displacements of the sea ice edge.

Ebert and McBride (2000) examined the position error of the contiguous rain area in weather forecasts. They determined the error vector from a total squared error minimization method when shifting the forecasted rain region to match the corresponding

observations. Their preference of applying an error minimization algorithm rather than an MCC approach was motivated by the former having better representations of displacement of rainfall maxima. An object-based approach with a focus on assessing the quality of forecasts for highly localized and episodic phenomena was introduced by Davis et al. (2006). Like in Ebert and McBride (2000), their focus when evaluating displacement is on the objects' centroid. Displacement of the perimeter of the contiguous rain area was not addressed in either investigation.

We begin this study by presenting a new algorithm for assessing the quality of representations of the sea ice edge by comparing results for two different data sets. We examine displacements over time of the sea-ice edge, and compare model results for displacement distances versus observed displacements. The method is described in Sect. 2 with an idealized case study. In Sect. 3 we apply the algorithm to analyze displacements of the sea ice edge in the Barents Sea. Finally, we provide our concluding remarks in Sect. 4. Technical details and extensions are provided in two appendices.

## 2   Methods

In order to illustrate the validation metrics that are introduced in this section, a set of idealized ice edges is introduced, as depicted in Fig. 1. The domain is divided into $1000 \times 500$ square grid cells, and we set the length of the side of a grid cell to 1. Denote the curve that separates regions with binary values 0 and 1 as an edge curve, and let $L^O(t)$ and $L^M(t)$ denote observed and modeled edges, respectively, at time $t$. Idealized examples with edges for $L^O$ and $L^M$ at two different times, $t_0$

and $t_0 + \Delta t$, are displayed. In the context of forecasting, $L^M(t_0)$ may be taken to represent the model initialization at $t_0$ and $L^M(t_0 + \Delta t)$ is then the forecast at a temporal range of $\Delta t$. The other binary fields can represent observations at the same times.

### 2.1   Validation metrics for ice edge displacement in one data set

We aim at defining metrics that describe differences in maximum sea ice expansion from sea ice edge displacements between

two data sets. In order to do so, we must first introduce a quantity that properly measures the maximum displacement in one data set. Here a definition is provided which is a gridded, signed, one-sided variation of the Hausdorff distance (Dukhovskoy et al., 2015).

For the remainder of this investigation we will take the binary fields to be representations of sea ice, with values assigned to 0 and 1 for conditions of no ice and ice, respectively. We will here associate the presence of ice (value 1) with sea ice

concentration $c$ exceeding $c_{edge} = 0.15$. In a gridded representation the ice edge can then be taken to be constituted by the grid

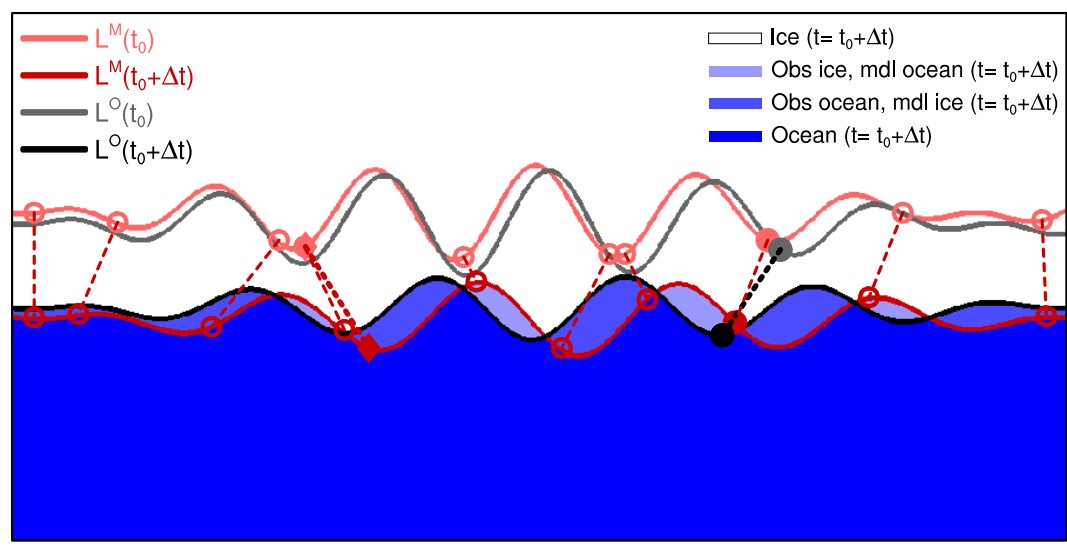

**Figure 1.** Binary fields with values of 1 (ice) and 0 (no ice/ocean) are displayed by white and blue color shading, respectively. Light shades of blue indicate regions with a non-overlapping ice cover, as indicated by the color legend. The derived modeled and observed ice edges $L^M$ and $L^O$ at $t=t_0+\Delta t$ are drawn as red and black curves, respectively. The corresponding ice edges that are taken to represent the situation at $t_0$ are drawn as light red and gray curves. The full black circle indicates the position on the observed ice edge at $t_0+\Delta t$ which has the largest distance to the ice edge at $t_0$ (the full gray circle). The largest displacement of the model ice edge is marked by full diamonds. The full red circle is the position along the model ice edge at $t=t_0+\Delta t$ closest to the full black circle, while the full light red circle is the position of the observed ice edge at $t_0$ closest to the full red circle. All dashed lines represents displacements as defined by Eq. (3). Open circles indicate a random selection of displacement positions for the model results, see the text for details.

cells $e = [i, j]$ that meet the condition

$$c[i,j] \geq c_{edge} \quad \wedge \quad \min\big(c[i-1,j], c[i+1,j], c[i,j-1], c[i,j+1]\big) < c_{edge} \tag{1}$$

where $\wedge$ is the logical AND operator. Denoting the $N(t)$ grid cells that satisfy this condition for time $t$ by $e_1^{(t)}, e_2^{(t)}, \ldots, e_N^{(t)}$ the ice edge for time $t$ is then the curve

$$L(t) = \{e_1^{(t)}, e_2^{(t)}, \ldots, e_{N(t)}^{(t)}\} \tag{2}$$

This follows the algorithm presented in Melsom et al. (2019). Let $L(t_0), L(t_0 + \Delta t)$ denote the sea ice edges at times $t_0$ and $t_0 + \Delta t$, respectively. Furthermore, let $d_n^{\Delta t}$ be the displacement distance between a grid cell $e_n^{(t_0+\Delta t)}$ in $L(t_0 + \Delta t)$ and curve $L(t_0)$, i.e.

$$d_n^{\Delta t} = s_n \min ||e_n^{(t_0+\Delta t)} - L(t_0)|| \tag{3}$$

Here, $||z||$ is the Euclidean distance of $z$ and the minimum is considered for the distances between grid cell $e_n^{(t_0+\Delta t)}$ and each of the grid cells belonging to $L(t_0)$. Furthermore, $s_n$ is +1 or -1 when $e_n^{(t_0+\Delta t)}$ is on the no ice or ice side of $L(t_0)$, respectively,

i.e.

$$
s_n = \begin{cases} -1 & \text{if } c[e_n^{(t_0+\Delta t)}](t_0) \geq c_{edge} \\ +1 & \text{if } c[e_n^{(t_0+\Delta t)}](t_0) < c_{edge} \end{cases}
\tag{4}
$$

where $c[e_n^{(t_0+\Delta t)}](t_0)$ denotes the sea ice concentration at the time $t_0$.

Figure 1 shows an idealized example where a modeled and an observed sea ice edge are displaced. Presently, we consider metrics for one product. To illustrate, focus here on the ice edges derived from the model product (light red and red lines). The length of dashed lines in Fig. 1 then correspond to model displacements $\min ||e_n^{(t_0+\Delta t)} - L(t_0)||$ for selected cells $e_n^{(t_0+\Delta t)}$. We introduce the maximum expansion displacement as

$$
d_{max}^{\Delta t} = \max(d_n^{\Delta t}) \quad , \quad n = 1, 2, \ldots, N(t_0 + \Delta t)
\tag{5}
$$

Note that the definition of the sign $s$ in Eq. (3) has been chosen so that Eq. (5) will return the largest positive value among $d_n^{\Delta t}$. If all values of $d_n^{\Delta t}$ are negative, the result is the negative value distance with the lowest magnitude. The definition of $s$ was designed so that $d_{max}^{\Delta t}$ will represent the displacement of the largest sea ice advance from $L(t_0)$ to $L(t_0 + \Delta t)$.

If we briefly introduce $d_m^{-\Delta t}$ as the shortest distance from a grid cell $e_m$ of $L(t_0)$ to the curve $L(t_0 + \Delta t)$, we note that the Hausdorff distance $d_H$ between curves $L(t_0 + \Delta t)$ and $L(t_0)$ is

$$
d_H = max(|d_n^{\Delta t}|, |d_m^{-\Delta t}|)
\tag{6}
$$

Here, $d_H$ is symmetric with respect to the distance when swapping the two compared data sets (usually an observed and a modeled feature). The corresponding distance measure introduced in Eq. (5), on the other hand, is asymmetric by design: Distances are compute from the grid cells of the updated ice edge at $t_0 + \Delta t$ to the ice edge curve from the initialization at $t_0$.

For the various quantities we reference model results and observations by superscripts $M$ and $O$. For the set of binary fields depicted in Fig. 1, we find that $d_{max}^{M;\Delta t} = 113.2$ (the distance between the red and light red diamonds in the figure), while $d_{max}^{O;\Delta t} = 97.9$ (the distance between the black and gray full circles).

The maximum distance in Eq. (5) provides a single measure to examine the ice-edge displacement. However, it can be more informative to analyze the whole distribution of the displacements $d_n^{\Delta t}$ defined by Eq. (3), rather than their maximum only. This can be done by inspecting a histogram of the displacements $d_n^{\Delta t}$ (Fig. 2). Another option is to present the cumulative probability distribution of $d_n^{\Delta t}$ (Fig. 3).

To avoid inflating the sample size beyond its degrees of freedom, displacement results can be subsampled at the spatial decorrelation length along the ice edge . For the distribution of $d_n^{\Delta t}$ a proper decorrelation length can be computed if the edge cells $e_n^{(t_0+\Delta t)}$ are in sequence along $L(t_0 + \Delta t)$. A detailed description for this procedure is given in Appendix A. In Sect. 3.3 we will present results for the time series $d_{max}^{\Delta t}(t)$ derived from subsampling based on decorrelation.

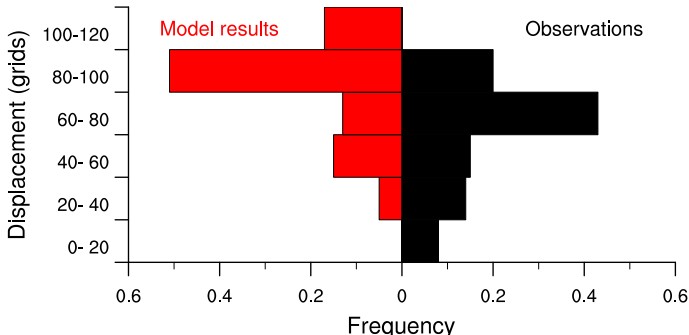

**Figure 2.** Histograms for the distribution of displacement distances computed from Eq. (5) for the ice edges displayed in Fig. 1. The mean displacement distances for model results and observations are 82 grids and 61 grids, respectively. The corresponding median values are 88 grids and 70 grids, respectively.

## 2.2 Comparison of the displacements of modeled and observed ice edges


In the previous section we focused on metrics which describe the displacement of a single (modeled or observed) sea ice edge. In this section we extend these to assess the differences in the displacements of the modeled versus observed ice edge. For this purpose, binary fields that are taken to represent observations as well as model results are introduced, as displayed in Fig. 1.

We aim to compare model results for displacement distances with the corresponding displacements from observational data.

Here, we discuss results for the idealized case depicted in Fig. 1.

In Fig. 2 the histograms for all displacements $d_n^{\Delta t}$ are shown, with the histograms to the left and right representing model results and observations, respectively. The results have been binned into categories that each span distances of 20 grid cell units. Displacement distances are not evenly distributed, as they have about half of the distance values in one of six categories, and this specific category is shifted by one category from observations to model results.

Next, the cumulative distributions of displacement distances are displayed in Fig. 3. Here, the results were subsampled by the decorrelation length along the edge curves prior to the analysis. We find that the model displacements are shifted approximately 20 grid units higher for the entire distribution, as the two curves are nearly parallel.

Hence, Fig.s 2 and 3 both reflect the distributions' near uniform shift of 20 grid units, and show that this shift is not qualitatively impacted from subsampling at the decorrelation length.

From the perspective of an observer, a useful attribute is the quality of the forecasted maximum displacement of the binary field, over the forecast period. A simple metric which provides this type of information is the difference in the maximum displacement as given by Eq. (5), i.e.

$$\Delta d_{max}^{\Delta t} = d_{max}^{M;\Delta t} - d_{max}^{O;\Delta t} \tag{7}$$

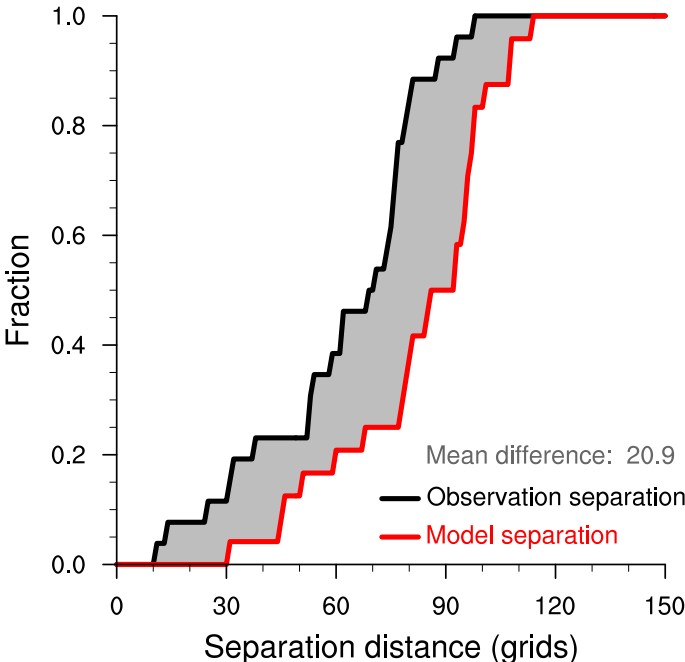

**Figure 3.** Cumulative distributions of the separation (ice edge displacement) distances from $t_0$ to $t_0 + \Delta t$, for model results (red curve) and observations (black curve). Shown here are results for the idealized example displayed in Fig. 1, with distances subsampled at intervals of the decorrelation lengths, which are 42 and 38 grid cells along the ice edge for the model results and observations, respectively. The mean separation distance difference for the present subsample of ice edge grid cells is the integral of the area between the curves, here displayed by gray shading. In this case, the mean difference is 20.9 in grid cell units, with larger displacement values in model results than from observations.

where $d_{max}^{O;\Delta t}$ is computed from observed ice edges at $t_0$ and $t_0 + \Delta t$ (black and gray curves in Fig. 1, respectively), and $d_{max}^{M;\Delta t}$ is computed from the corresponding model results. For the results in the idealized example that was introduced in Sect. 2.1, the model is over-estimating the maximum displacement, by $\Delta d_{max}^{\Delta t} = 15.3$ grid cell units.

A similar quantity that provides local information is the local difference in displacement of the model ice edge in proximity of the maximum displacement found in the observations. Let $e_0^{O;(t_0+\Delta t)}$ be the position in $L^O(t_0 + \Delta t)$ to which the maximum edge displacement is found in the observations. Then, determine $\epsilon_0^{M;(t_0+\Delta t)}$, the model edge grid cell positioned closest to $e_0^{O;(t_0+\Delta t)}$ at the same time. In Fig. 1, the positions $e_0^{O;(t_0+\Delta t)}$ and $\epsilon_0^{M;(t_0+\Delta t)}$ are indicated by the full black and full red circles, respectively. Following Eq. (3) the corresponding local edge displacement in the model results is

$$\delta_0^{M;\Delta t} = s \min ||\epsilon_0^{M;(t_0+\Delta t)} - L^M(t_0)|| \tag{8}$$

For the idealized example, we find that $\delta_0^{M;\Delta t} = 83.9$. The local difference in displacement between model and observations, with reference to the position $e_0^{O(t_0+\Delta t)}$, becomes

$$\Delta \delta_{max}^{\Delta t} = \delta_0^{M;\Delta t} - d_{max}^{O;\Delta t} \tag{9}$$

We recall from Sect. 2.1 that $d_{max}^{O;\Delta t} = 97.9$ so, for the idealized example we have $\Delta\delta_{max}^{\Delta t} = -14$, i.e. a local underestimation of the displacement in the model results.

One aspect which is not disclosed by the metrics introduced thus far, is to what degree forecasts manage to reproduce the geographical location of the observed maximum displacements. In order to examine such a relation, we first compute the decorrelation length of displacements given by Eq. (3). If we denote this grid distance by $\Delta n$, we restrict the analysis of grid cells and corresponding displacements to

$$\{\ldots, \epsilon_{0-2\Delta n}^{M}, \epsilon_{0-\Delta n}^{M}, \quad \epsilon_{0+\Delta n}^{M}, \epsilon_{0+2\Delta n}^{M}, \ldots\}(t_0 + \Delta t), \tag{10}$$

$$\{\ldots, \delta_{0-2\Delta n}^{M;\Delta t}, \delta_{0-\Delta n}^{M;\Delta t}, \quad \delta_{0+\Delta n}^{M;\Delta t}, \delta_{0+2\Delta n}^{M;\Delta t}, \ldots\} \tag{11}$$

respectively, limited by the first and last cells along the curve $L^M(t_0 + \Delta t)$. Next, we construct bins analogously to the method used for producing rank histograms (Talagrand diagrams) for ensemble forecasts (Hamill, 2001; Talagrand et al., 1997; Anderson, 1996): First, distances listed in Eq. (11) are sorted by increasing values, and then bins are introduced for values smaller than the minimum distance, the intervals between the sorted distances, and for values larger than the maximum distance. The bin placement of $\delta_0^{M;\Delta t}$ then gives the rank of this displacement. With a perfect model, the maximum of $\delta^{M;\Delta t}$ will occur at the geographical position corresponding to the maximum displacement in the observations ($d_{max}^{O;\Delta t}$). This will place $\delta_0^{M;\Delta t}$ in the highest (rightmost) bin. A flat histogram indicates a model with no skill, since each bin is equiprobable for random draws.

In the present idealized example we find that $\Delta n = 42$, and the rank of $\delta_0^{M;\Delta t}$ in the 24 resulting bins is 9. When multiple forecasts are examined, the decorrelation length will generally change, as will the length of the edges. Thus, we randomly subsample a fixed number of intervals from Eq. (11), so that the number bins is equal across different cases and results can be aggregated.

For the idealized example, a set of nine randomly subsampled edge positions from those given by Eq. (10) for the model results at $t = t_0 + \Delta t$ is displayed by open circles in Fig. 1. For this particular case, in the range from 1 to 10 the rank of the displacement $\delta_0^{M;\Delta t}$ is 3. So, in this idealized, synthetic example, the model exhibits a poor positioning of the maximum observed displacement.

## 3 Application of the new validation method

### 3.1 Description of sea ice data sets

To illustrate the methodology introduced in Sect. 2, we examine model results from a coupled ocean – sea ice model, and compare with relevant observational data. The model results are taken from the SVIM hindcast archive (SVIM, 2015). For the present illustrative purpose we limit the analysis to the two year period from 1 January 2000 to 31 December 2001. Results are available as daily means on the model configuration's native 4 km stereographic grid projection (Lien et al., 2013).

The ocean module of the coupled model used for the regional simulation is the Regional Ocean Modeling System (ROMS), described in Haidvogel et al. (2008) and references therein. The sea ice module was developed by Budgell (2005). The ice

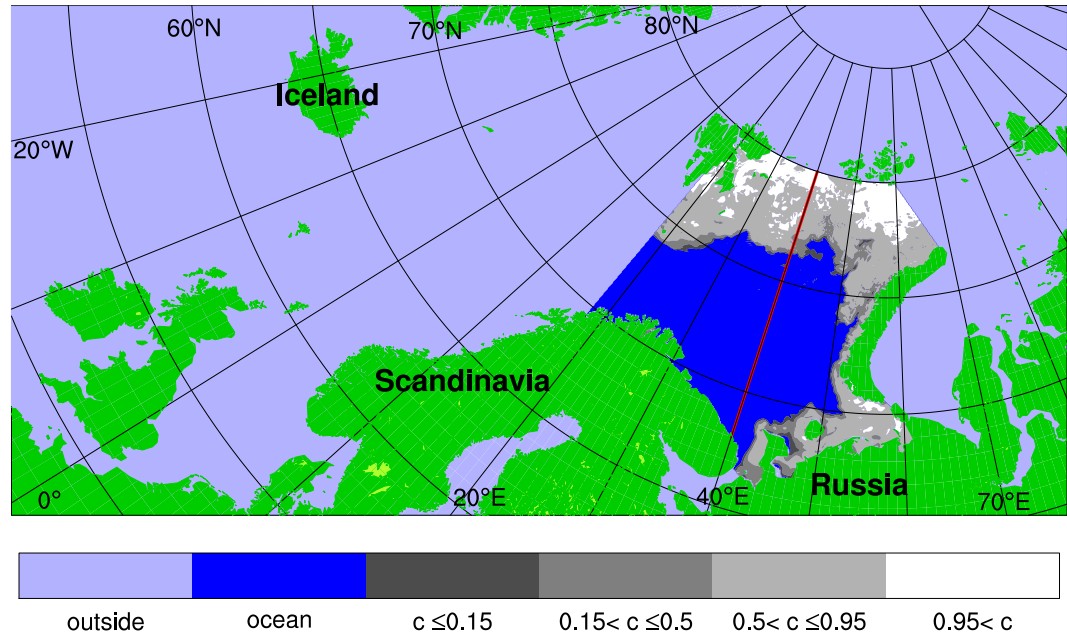

**Figure 4.** Map of the full SVIM simulation domain. The Barents Sea analysis region in the present study is shown as a highlighted region where a sample sea ice concentration distribution is depicted. The 40°E meridian which will subsequently be used for dividing the domain into two parts, is displayed by the red curve. The shading of ice concentration values is given in the label bar, where c is in the sea ice concentration fraction. This sample shows the model results for 15 April 2000, with the horizontal resolution from the SVIM experiment.

model dynamics are based on the elastic-viscous-plastic (EVP) rheology after Hunke and Dukowicz (1997) and Hunke (2001), and the ice thermodynamics are based on Mellor and Kantha (1989) and Hakkinen and Mellor (1992).

The model results for sea ice concentration are somewhat noisy on the grid cell scale, owing to the dispersiveness of the nu-
160 merical scheme. In some regions, the grid cells that constitute the ice edge as defined by Eq. (1) can then appear as a mesh-like collection of cells. In order to reduce the impact of this issue, we applied the second order checkerboard suppression algorithm (Li et al., 2001) to the model results before conducting the present analysis. The sea ice concentrations from observations does not suffer from this type of noise, thus such an algorithm was not applied to the observational data set.

We compare model results with observations from the Arctic Ocean Sea Ice Concentration Charts *Svalbard* which is a multi-
165 sensor data set that uses data from Synthetic Aperture Radar (SAR) instruments as its primary source of information (WMO, 2017). This observational data set will be referred to as the ice chart data hereafter.

The ice chart data cover the northern Nordic Seas, the Barents Sea and adjacent ocean regions. The ice chart data deviates from a passive microwave product in this region, particularly in the final months of the melting season (e.g. Sect. 6 in Melsom et al. (2019)). Hence, the true sea ice extent is unknown.
The ice chart data are available on a stereographic grid projection with a resolution of 1 km. Data availability is restricted to working days. During a regular week, we then have four days with 24 h displacement results. The data set is also slightly

reduced due to holidays, and a total of 354 days with 24 h ice edge displacement results were available from the present two year period.

The present study will be restricted to results and data for the Barents Sea. The SVIM simulation domain is displayed in Fig. 4, where the Barents Sea analysis region is highlighted. Ice chart results are integrated onto the SVIM domain using a mass conserving Riemann integral approach. All grid cells inside the Barents Sea region which lack proper values (usually due to the presence of land) in at least one of the data sets are masked prior to the analysis. The analysis region is then constituted by 80.399 wet grid cells, which represent an area of $1.29 \cdot 10^6$ km$^2$.

## 3.2 Open boundaries and coasts

Situations may arise when the algorithm described in Sect. 2.1 gives rise to an unrealistic representation of displacement distances, and this may particularly affect the maximum value as defined by Eq. (5) and quantities that depend on this definition. We can illustrate this issue by inspecting the change in the ice edge position from 23 October 2001 to 24 October 2001, which is an extreme case in the present context. The evolution of the model results over this 24 h period is displayed in Fig. 5.

The example demonstrates that in such cases, the general algorithm in Sect. 2.1 mismatches ice edge grid cells when the algorithm identify ice edge motion incorrectly. A straightforward implemention of the algorithm in Sect. 2.1 leads to a maximum sea ice displacement of 285 km as given by the thick black line close to the subdomain's northern border.

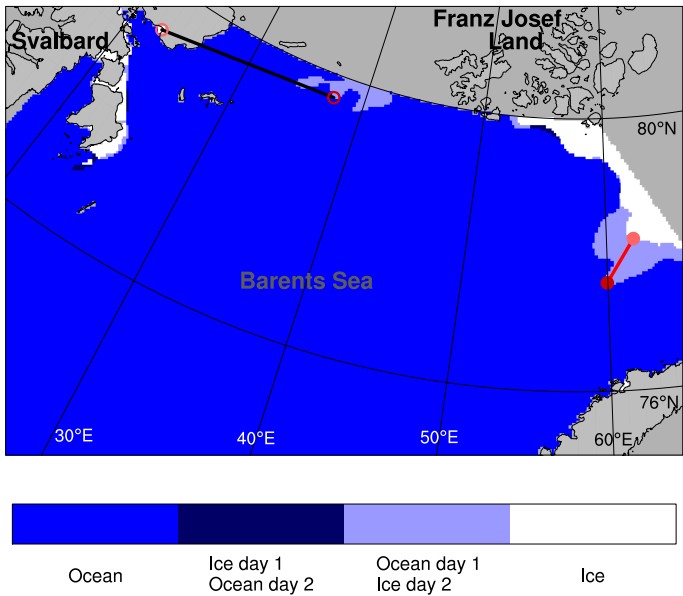

**Figure 5.** Sample scene displaying the changes in model sea ice extent from 23 October 2001 (day 1) to 24 October 2001 (day 2). The black line indicates the maximum displacement distance ($d_{max}^{\Delta t}$, given by Eq. (5) with the original algorithm, while the red line shows the result when grid cells along the open boundaries and coastlines are included ($\widetilde{L}(t_0)$ from Eq. (B7)). The color coding is given by the label bar, and note that only the northern part of the Barents Sea analysis region is displayed.

This is clearly a case where sea ice is advected into the domain across an open boundary, and the algorithm must be modified in order to avoid misinterpretations of results for displacement distances. This can be done by including grid cells along the open boundary as potentially being neighbours to the sea ice edge. Corresponding modifications to the displacement distance algorithm are introduced in Appendix B. The maximum distance using the modified algorithm becomes 79 km (red line). It must be noted that if the ice is advected into the domain, the distances associated with such a displacement will be underestimated, since the real position of the ice edge outside of the analysis domain at $t_0$ is unknown.

Situations where unrealistic representations for displacements may also arise when ice freezes along the coast, e.g. due to colder air in the vicinity of continents, or less salty water masses close to the coastline. This issue may be treated analogously to advection across an open boundary, see Appendix B for details.

## 3.3 Validation results

We first examine the distribution of daily maximum ice edge displacements. From Fig. 4 we note that this examination will be performed for a domain in which advection of sea ice across the open boundaries is relevant, as well as freezing along coastlines of the continent and archipelagos. To address this issue, the analysis based on the algorithms in Sect. 2 will be extended following the outline in Sect. 3.2, and detailed in Appendix B.

In Fig. 6 results from the 354 days with 24 h maximum displacements from both data sets are displayed. We note that about 2/3 of the maximum displacements in model results are in the range 10 – 30 km. The corresponding distribution of results from the ice chart data has two maxima, one for the range 20 – 40 km which accounts for nearly half of the cases, and a secondary maximum for short (0 – 10 km) maximum displacements. The medians of the daily maximum displacement distances are 23 km and 32 km for the SVIM results and the ice chart data, respectively. We also note that the distribution frequencies for the two year period change only moderately when the adjustments for open ocean boundaries and coastlines that were described in Sect. 3.2 are included in the analysis.

A conclusion that can be drawn from these results is that the largest expansions of sea ice extent in the model (SVIM) results are underestimations when compared with observations (ice chart data). This is generally the case, as the SVIM median is in the range 20 – 30 km, while the median of the ice chart data is in the range 30 – 40 km. The underestimation is also seen for extreme cases, as the frequency of maximum expansion exceeding 60 km is about five times as high for the ice chart data.

In order to examine the degree to which SVIM results reproduce the geographical location of the observed maximum displacement, we apply the ranking method described in Sect. 2.2. We consider a fix number of 10 bins for the present investigation. Hence, for each set of 24 h results for displacement distances, nine values are randomly selected from the displacements in Eq. (11). Moreover, the requirement of at least nine additional ice edge positions separated by the decorrelation length scale restricts the cases that can be considered in this analysis. From the full set of 354 cases with 24 h displacement results, 235 cases could then be kept in the analysis of ranked displacements, as these cases met the requirement of at least 10 statistically independent displacement distances. The size of the set of independent values is restricted by the temporally varying degrees of freedom, as given by the ice edge length and the decorrelation length scale.

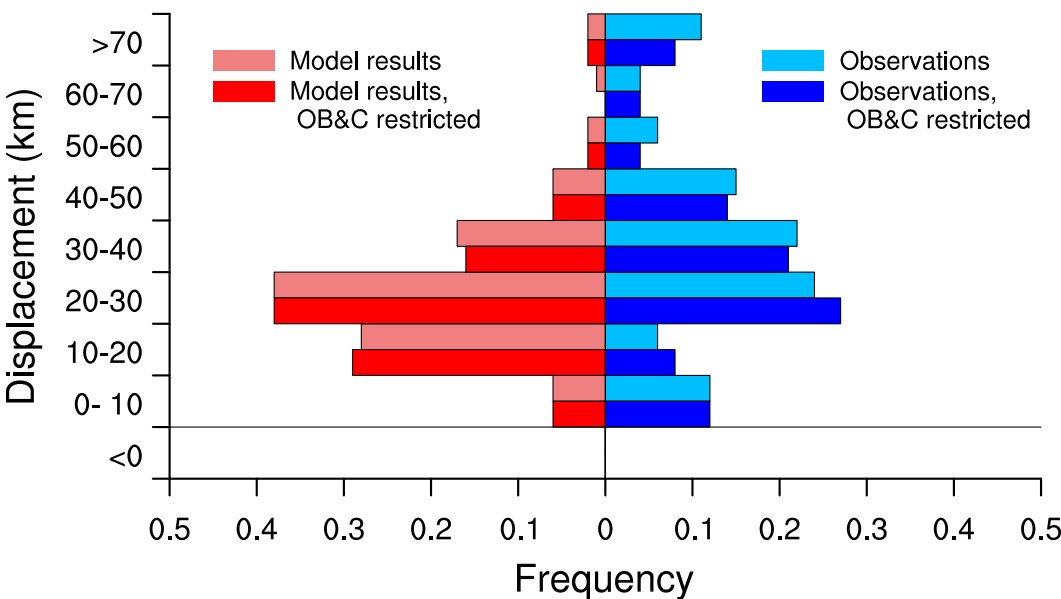

**Figure 6.** Histograms for the distribution of daily maximum displacement distances $\Delta d_{max}^{\Delta t}$, defined by Eq. (5). Horizontal bars pointing left and right correspond to results from SVIM model simulation and ice chart data, respectively. Light colored bars display results from the original algorithm in Sect. 2.1, while bars with regular colors (labeled "OB&C restricted") result after the extension in Appendix B for open boundaries and coastlines is applied. Results from 354 days of 24h edge displacements have been analyzed, see the text for further details.

The resulting frequency distribution for each of the ten ranks is displayed as gray vertical bars in Fig. 7(a), with rank values from 0 to 9. The highest rank (9) results when the model displacement close to the site with maximum displacement in the observations (the reference displacement, $\delta_0^{M;\Delta t}$) is larger than all displacements from the nine subsampled ice edge positions. The next rank (8) corresponds to cases where one and only one of the subsampled positions have a larger displacement than the reference displacement, and so on. In other words, high ranks indicate situations in which the position of the maximum

displacement is described with a relatively high quality.

     The histogram in Fig. 7(a) has nearly twice as many entries in ranks $5-9$ than ranks $0-4$. This mode for higher ranks indicate some skill for results from the SVIM archive in detecting the location of the maximum displacements in the observations. The average rank in the present analysis is 5.48. For a random distribution of 235 integer numbers in the range $0-9$ the $0.5^{th}$ and $99.5^{th}$ percentiles of the average rank are 4.02 and 4.98, respectively. Thus, the analysis reveals that while the model results

are far from perfect, the average rank of 5.48 is significantly higher than results from random spatial distributions of ice edge displacements. We have also applied the more traditional $\chi^2$-test for rank flatness Wilks (2019), and find that for the present histogram we have $\chi^2 = 52$. This value is nearly twice the magnitude of $\chi_{crit}^2$ when this is set to reject the null-hypothesis of a flat distribution at an $\alpha$ level of 0.001.

     We supplement this analysis by dividing the region into two subdomains, separated by the $40°E$ meridian, as indicated by

the red line in Fig. 4. In order to retain the majority of the days in the analysis, the number of randomly chosen displacements

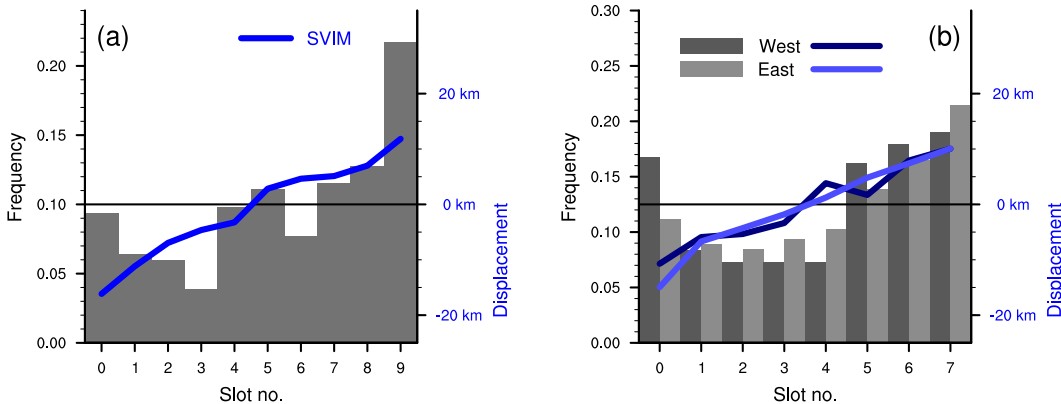

**Figure 7.** Rank histogram for model results for the local ice edge displacement corresponding to the position of the maximum observed displacement. (a): Sets of nine alternative model displacements were derived for each of 235 days with 24 h displacements results. The nine displacement values were ordered from lowest to highest, and the local displacement was given a rank from the slot in which this value belonged, see the text for details. The blue curve shows the average local model displacement distances for results belonging to each of the ranks, with negative numbers corresponding to local sea ice retreat in the model results. The average maximum observed displacement is 39 km. (b): Results obtained for a subdivision as indicated by the red line in Fig. 4. Here, sets of seven alternative displacements results were derived for each subdomain. Then, results were available for 179 days and 224 days for the western and eastern subdomain, respectively. The average maximum observed displacements are 36 km in both subdomains. The use of gray shading and line colors for the subdomains are indicated by the inset label. The y-axes in (a) and (b) have been specified so that the black horizontal 0-axes lines for the righ-side axes correspond to the frequency level of a flat distribution of frequencies (left-side y-axes).

was reduced to seven values. This was due to the reduced degrees of freedom when the same decorrelation length scale was applied in smaller domains, with shorter ice edges.

The results for the eastern and western Barents Sea subdomains are displayed in Fig. 7(b). Contrasts between the frequency distributions between panels (a) and (b) arise for several reasons. First, the domain split leads to a set of two time series

of maximum displacements where the maxima from the full domain will be distributed between the two subdomain time series, and new maxima are introduced for the alternative subdomain. Next, the separation line between the two subdomains is manifested in the analysis as a new, shared open boundary. Hence, the shapes of the subdomain distributions may to some degree deviate from the full domain distribution. We note that for the present analysis, the result is that the distribution peak in ranks 4 and 5 from the full domain case disappears in the resulting rank frequencies for the subdomains.

The average ranks of the model displacements corresponding to the largest observed displacements are 3.95 and 4.13 in the western and eastern subdomain, respectively. The ranges spanned between the 0.5th and 99.5th percentiles for random distributions become [3.06,3.94] and [3.11,3.89] for the western (179 days) and eastern (224 days) subdomain, respectively.

## 4 Concluding remarks

In this study we present a new algorithm for examination of the displacement of the edge (or the front) of a binary field. From this algorithm we can compute properties for displacement distances. Next, methods that compare such properties are introduced in order for the quality of edge displacement results from a model to be quantified. These methods are e.g. relevant when assessing the quality of forecasted displacements of the edge. The results from these methods expand on existing validation metrics such as e.g. the Integrated Ice-Edge Error (Goessling et al., 2016) and the various ice edge metrics considered by Melsom et al. (2019): The methods presented here provide summary statistics for the quality of model results for ice edge displacements in the presence of an expanding sea ice cover, as exemplified by Fig. 6 and Fig. 7, that are not provided with existing metrics. Such quality assessments are of high relevance for planned or ongoing site specific activities in regions which can potentially become ice infested.

The present study has been framed in the context of results for displacements of the sea ice edge. Thus, the investigation in Sect. 3 was based on data for the sea ice edge from satellite observations, and simulation results from a coupled ocean – sea ice model. However, the algorithm that was introduced in Sect. 2 can be applied to the displacement of the perimeter of any property that can be represented by a continuous binary field. Stratiform precipitation is an example of another property for which the methods presented here are relevant.

Note that we have used the term *displacement* rather than *advection*. The reason for this is that displacements need not be purely of an advective nature. In the case of sea ice, the displacement of the initial edge will generally be affected by freezing or melting along the perimeter of the sea ice extent. Analogously, displacement of the area affected by stratiform precipitation can be affected by new condensation or partial depletion of the cloud.

As demonstrated in the example depicted in Fig. 5, the original algorithm described in Sect. 2.1 and 2.2 may yield results that represent other aspects than true displacements. Here, we have amended situations in which the sea ice enters a limited area domain across an open model domain boundary, and situations where freezing takes place next to a physical boundary (the coast). Modifications of the algorithm which include distances from ice edges to coasts and open boundaries are described in Sect. 3.2 and detailed in Appendix B. This approach eliminates unphysical edge displacement distance values, as revealed from the sample situation in Fig. 5.

However, there may be other issues that can distort results that are produced by the analysis presented here. One example is cases where features are seen to arise seemingly spontaneous from one time of analysis to another: The algorithm in Sect. 2 can e.g., if applied to precipitation data, give rise to unrealistic results for displacements when convective precipitation cells develop.

Results from the algorithms that are introduced in the present study give valuable information regarding the changing extent of sea ice, and how well the displacements of the observed and modeled sea ice edges agree. These algorithms have proven to provide simple, yet robust and informative assessments for the quality of ice edge forecasts both with respect to the largest displacements from one time to another as well as with respect to the reproduction of the geographical position where the largest displacement occurs.

*Code and data availability.* The idealized distributions of concentrations that are depicted in Fig. 1, and source code for computing results displayed in Fig. 2 and Fig. 3, are available from https://doi.org/10.5281/zenodo.4545686 (Melsom, 2021). Model results that were analyzed in Sect. 3 are available from https://archive.norstore.no/pages/public/datasetDetail.jsf?id=10.11582/2015.00014 (SVIM, 2015), while the observations are available from https://thredds.met.no/thredds/catalog/arcticdata/met.no/iceChartSat/catalog.html.

## Appendix A: Decorrelation length of displacements

Assume that a we have a set of $N$ edge grid cells $e_n$ (i.e. satisfying Eq. (1)) that form a curve

$$L = \{e_1, e_2, \ldots, e_N\} \tag{A1}$$

where $L$ is continuous in the sense that grid cells $e_n$ and $e_{n+1}$ are neighbors. Furthermore, associate displacement distances $d_n$ to each $e_n$ as defined in Sect. 2.1. Then, the spatial autocorrelation of displacements can be estimated using a sample Pearson correlation coefficient approach:

$$r(\eta) = \sum_{n=1}^{N-\eta} [(d_n - \overline{d_n})(d_{n+\eta} - \overline{d_{n+\eta}})] \quad \Big/ \quad \left[ \sum_{n=1}^{N-\eta} (d_n - \overline{d_n})^2 \sum_{n=1}^{N-\eta} (d_{n+\eta} - \overline{d_{n+\eta}})^2 \right]^{1/2} \tag{A2}$$

We have $r(0) = 1$ and we define the decorrelation length of the displacements, $\Delta n$, as

$$\Delta n = \min_{\forall \eta}(\eta \mid r(\eta) < 1/\mathrm{e}) \tag{A3}$$

where e is Euler's number. If, for a given time, the ice edge is discontinuous, each continuous curve segment is treated separately, and the weighted mean value of the results for $\Delta n$ from each segment is used. In that case, weights are applied according to the number of edge grid cells in each curve segment.

## Appendix B: Open boundaries and coasts

As discussed by Melsom et al. (2019), open boundaries and coastlines can potentially have significant impacts on the results for the metrics for the position of the sea ice edge. Here, we introduce a method which will give more physical meaningful results if ice either freezes near a coastline, or enters into a domain across an open boundary. Moreover, the method affects the results modestly or not at all for an edge that is displaced inside of the domain.

First, set the open boundary grid lines to

$$L^{OB}(t_0) = \{e_{1_{OB}}, e_{2_{OB}}, \ldots, e_{N_{OB}}\} \quad , \quad c[e_{n_{OB}}](t_0) < c_{edge} \tag{B1}$$

where $e_{n_{OB}}$ is any ocean grid cell along the boundary of the domain which was on the open ocean side of the ice edge at $t = t_0$. Then $L(t_0)$ in Eq. (3) can be replaced by

$$\widetilde{L}(t_0) = L(t_0) \cup L^{OB}(t_0) \tag{B2}$$

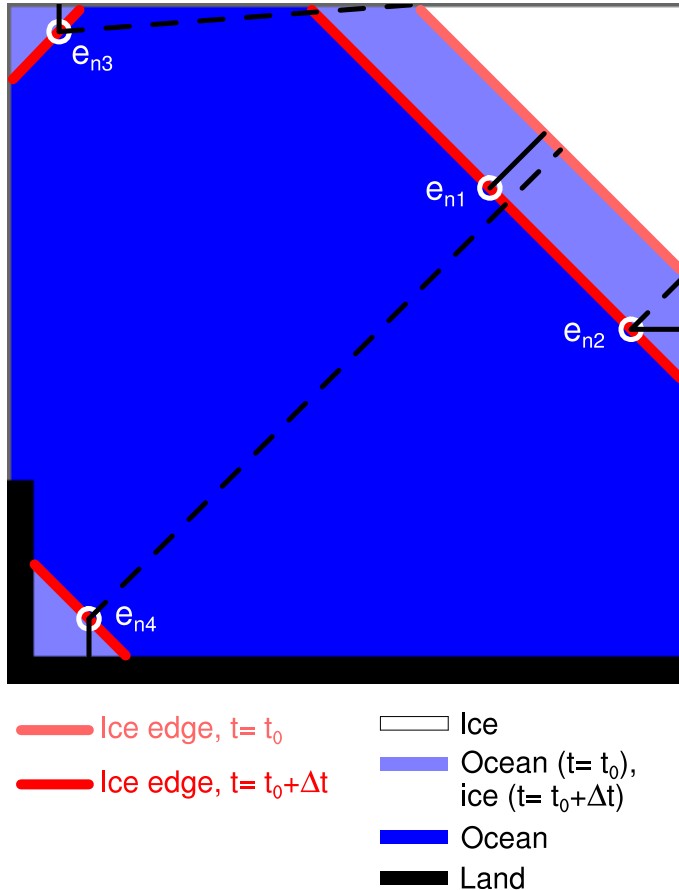

**Figure B1.** Binary fields with values of 1 (ice) and 0 (no ice/ocean) are displayed by white and blue color shading, respectively. Land is indicated as a black region. Light shades of blue indicate regions with a non-overlapping ice cover, as indicated by the inset color legend. Open boundary grid cells are depicted as gray lines. Ice edges for $t_0$ and $t_0 + \Delta t$ are drawn as light red and red lines, respectively. Dashed black lines show the edge displacements as defined in Sect. 2.1, for a selection of labeled ice edge grid cells from $t_0 + \Delta t$, marked by white circles. Full black lines display the displacements $\widetilde{d}$ that result from the modifications described in Appendix B, see Eq. (B8). For the set of grid cells that are highlighted here, only $e_{n1}$ is unaffected by the modified definitions.

and for the corresponding distances we introduce the notation $\widetilde{d}$, so Eq. (3) becomes

$$\widetilde{d_n}^{\Delta t} = \min ||e_n^{(t_0 + \Delta t)} - \widetilde{L}(t_0)|| \tag{B3}$$

where $e_n^{(t_0 + \Delta t)}$ is a grid cell on $L(t_0 + \Delta t)$, as before. The set of grid cells $e_n^{(t_0 + \Delta t)}$ is not affected, so the number of displacement distances considered in Eq. (5), $N(t_0 + \Delta t)$, is unchanged. Note that here, the additional curve $L^{OB}$ is only added to $L(t_0)$. Otherwise, if it was also added to $L(t_0 + \Delta t)$, the trivial score from perfect metching segments of edge curves would significantly impact the resuls as they are e.g. displayed in Fig. 2 and 3.

A sample grid cell to which the displacement distance is significantly affected by this modification, is displayed as $e_{n3}$ in Fig. B1. It must be noted that if the ice is imported into the domain, the distances $\widetilde{d}$ associated with such a displacement will be underestimated, since the real position of the ice edge outside of the analysis domain at $t_0$ is unknown. Moreover, for regular displacement of ice inside the domain, results will be affected slightly when occurring in the vicinity of the open boundary (e.g. $e_{n2}$ in Fig. B1).

Similarly, there can be cases where freezing of ice occurs along the coastline, e.g. as an effect of colder air in the vicinity of continents, or less salty water masses close to the coastline. This is another case where the algorithm above can yield grossly exaggerated displacement distances. Again, the problem can be alleviated by including additional grid lines.

Set the coastal grid lines as

$$L^C(t_0) = \{e_{1_C}, e_{2_C}, \ldots, e_{N_C}\} \quad , \quad c[e_{n_C}](t_0) < c_{edge} \tag{B4}$$

where $e_{n_C}$ is any ocean grid cell along the coastline which was ice free or had a sea ice concentration below $c_{edge}$ at $t = t_0$. Then $L(t_0)$ can be replaced by

$$\overline{L}(t_0) = L(t_0) \cup L^C(t_0) \tag{B5}$$

Here, Eq. (3) will be replaced by

$$\overline{d_n}^{\Delta t} = \min ||e_n^{(t_0 + \Delta t)} - \overline{L}(t_0)|| \tag{B6}$$

and the computation of the displacement distance to grid cell $e_{n4}$ in Fig. B1 is severely affected.

For a regional model, the typical situation is that there are both open boundaries and coastlines. In that case, we may combine the above modifications of the algorithm by adopting

$$\widetilde{\overline{L}}(t_0) = L(t_0) \cup L^{OB}(t_0) \cup L^C(t_0) \tag{B7}$$

and Eq. (3) can be written

$$\widetilde{\overline{d_n}}^{\Delta t} = \min ||e_n^{(t_0 + \Delta t)} - \widetilde{\overline{L}}(t_0)|| \tag{B8}$$

*Author contributions.*  Melsom performed the analysis in its entirety, produced all figures and wrote the article.

*Competing interests.*  The author declares that he has no conflict of interest.

*Acknowledgements.*  This study was performed within the Nansen Legacy project on behalf of the Norwegian Research Council, funded under contract no. 276730. Supporting funding was provided by the Copernicus Marine Environmental and Monitoring Service under Mercator

Océan from contract no. 2015/S 009-011301. Yvonne Gusdal is acknowledged for performing the hindcast simulation. All figures were made using the NCAR Command Language (NCL, 2017). Referee comments from two anonymous reviewers were especially helpful during the revision of this article. I am very grateful for the reviewers' efforts.

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
