# Peer review of "Edge displacement scores"

_The Cryosphere, 2020_

## Referee Comment (RC2)

**Reviewer replies**

Thank you to the author for responding to the comments in the review. I apologise that the original comments were somewhat too brief and wish they had been more constructive. Some of the lack of clarity is due to the novel nature of the method. Here I will try to address the specific points in the author's response:

1.  The case study in section 3 shows the method in practice and presents the category distribution for ice edge displacement distances for the model simulation and ice chart data under two different formulations. This section contains a lot of quantitative information, and it may be helpful to know how the end-user might translate the information, for example the distribution in table 2. The rank histogram in figure 5 and lines 200 to 205 give a more direct description of the range in the ability of the model has reproduced the position of max ice edge displacement. It would also be interesting to see if the ranks achieved by the model have a temporal or spatial pattern.

2.  The inclusion of the additional sentences is a good idea. It will clarify that while pre-existing methods provide summary statistics for the quality of model results, this method specifically targets ice edge displacements from an expanding ice cover. Alongside this method will be of a higher value (than summary statistics) for site specific activities as mentioned by the author.

3.  One additional comment: In line 203, it states that "For a random distribution of 235 integer numbers in the range 0 –9 the 0.005th and 0.995th percentiles are 4.015 and 4.985, respectively."  It seems to imply that a random distribution of 235 integers between 0 and 9 will have 4.015 and 4.985 as the 0.005th and 0.995th percentiles respectively. I tried to reproduce this with various integer distributions with mean at 4.5 and cut-off at 0 and 9, but was not able to do so. Is the distribution mentioned in the statement restricted to certain condition that I misunderstood?

I want to reiterate that this is a very interesting new method for measuring ice edge displacement and will potentially find good use in the community. I hope the comments provided will help the final paper be more accessible and clearer for all readers. Thank you.

---

## Referee Comment (RC3)

Revision CryoSphere-tc-2020-361
Author: A. Melson
Title: Edge displacement scores

**Overall general comments:**

This study introduces a new distance-based approach for assessing the quality of the representation of the displacement over time of the sea-ice edge. The methodology introduced is promising, however the article is yet not mature and need some major revisions for being acceptable for publication in the CryoSphere. Major revisions aim to improve the structure and exposition of the article, as well as some aspects of the verification method.

**Major revisions:**

1. **Section 2.1 needs major rewriting**. Since the author aims to maximize generalization, the description is too generic and therefore it is not clear (until you read section 3). A not exhaustive list of suggestions (for improvement) are:
   - State more explicitly and since the beginning (e.g. at lines 33, in the paragraph at lines 39-45, and then at line 47) that you aim to verify the displacement over time of the sea-ice edge, and compare the forecast versus observed displacements. (e.g. at line 47 the phrasing "the difference in maximum edge displacement between two products" can be mis-interpreted as the "distance between observed and forecasted ice-edges").
   - Do not use "product". You can use "dataset", "ice-edge" or other terms. You can as well refer to model (or forecast) and observation, explicitly.
   - The titles of Sections 2.1 and 2.2 should include "displacement" somewhere, e.g. "2.1 verification measures of the displacement of a single ice-edge" and "2.2. comparison of the displacements of forecast and observed ice-edges"
   - Lines 58-59: "$L^{(1)}, L^{(2)}$ denote the sea ice edges for two representations ... " is not clear at all: here **you need to state explicitly here that (1) indicate the time $t_0$ and (2) indicate the time $t_0+\Delta t$, and that you measure the ice-edge displacement that occurred between the times $t_0$ and $t_0+\Delta t$** (for either the forecast or the observation ice-edge). In fact, I suggest to change all the notation here (otherwise the reader will naturally associate 1 for forecast and 2 to observations), replacing (1) with ($t_0$) and (2) with ($t_0+\Delta t$), and then $d_n^{2:1}$ can become $d_n^{\Delta t}$.
   - The mathematical equations (3) and (4) need to be defined in a more mathematical rigorous way. Especially equation (4), the sign is not accounted for in the mathematical formula and there is a missing absolute value $|.|$.
   - Lines 62-63: the text is not precise; you need to state that in Eq. (3) you consider the minimum of all distances (between the single point in the edge at time $t_0+\Delta t$ and all points of the edge at time $t_0$). Similarly, for Eq(4) you need to state that you consider the max (over all n) of the $d_n^{\Delta t}$

- Lines 65: I believe this should state " … will return the largest *absolute* positive value … "
- Line 66: what happens if the $d_n^{2:1}$ are partially positive and partially negatives? Mention the canceling errors.

2. Section 2.1, Lines 70-75: **Table 1 (as well as Table 2** in Section 3) lists the frequencies of the histograms of the set of distances $d_n^{2:1}$ defned by Eq. (3), for the idealized model and observation ice-edges of figure 1. This is not clear from the text at line 75, nor from the caption (which I believe has a mistake, since it should refer to Eq. 3, and not Eq. 4) and the heading of the table is wrong (it should state "Frequencies" rather than "fraction of grid cells"). Here are my suggestions:
   - **Representing a distribution with a table is quite unconventional: I strongly recommend plotting the histograms of the distances $d_n^{2:1}$ instead**. (You can choose if to leave the table or eliminate it, but please add the histogram). Please show the histograms also for Table 2.
   - Please rephrase the caption ("category distribution of displacement distances" is not clear; this can simply be "distribution of ice-edge displacements").
   - Please rephrase lines 71-74: you can shrink it all as a couple of sentences such "The maximum distance in Eq. (4) provides a single measure to examine the ice-edge displacement. However, it can be more informative to analyse the whole distribution of the displacements $d_n^{2:1}$ defined by Eq. (3), rather than their maximum only. This can be done by analysing the histogram of the displacements (Figure HIST) and its corresponding frequencies (Table 1); the distribution of the displacements $d_n^{2:1}$ can as well be represented by their cumulative probability distribution (Figure 2)."
   - Please rephrase line 75: "the distribution of selected distance categories" is not clear.

3. The **decorrelation length**, used to sub-sample the ice-edge, is lightly mentioned at lines 78-80. It is thereafter used (e.g. caption of Figure 2, and then more heavily at lines 110-115), however a more thorough definition and how the author calculate Δn is missing in the article. Please add some text about it (maybe this can be done in an appendix).

4. Section 2.3 is too long, needs rewording, and need to be accompanied by a visual example. I strongly suggest to:
   a) summarize it in few sentences, such as in Melsom 2019, page 617, left column third paragraph ("A variant … "). In the re-wording it is important that you still retain the explanation at lines 138-139, where you correctly state that you expect the distances to be smaller when adding ocean open boundaries and coastal lines (because when adding these artificial "fixed" edges you automatically include in your verification some perfectly matched edges, aka trivial skill).
   b) **Include subsection 2.3 in Section 3**, after the data description and prior the results (aka after line 173 and before line 174). You can actually split section 3 in three subsections: 3a Sea-ice data description, 3b open boundaries and coasts, 3c verification results. In this fashion the reader has an immediate visual

example (figure 3) on your need of adding these fictitious boundaries (especially when considering an Arctic sub-region).

    c) The example presented at lines 180-187 and illustrated in figure 4 is excellent!

5. Give a new title to Section 3: Application of the new verification method on sea-ice forecast (this is not a "case study" ... also at line 211).

6. Lines 115-119: **rather than considering the ranks**, and a variable numbers of bins, I suggest using the **quantiles**, which have a fix range in [0,1] (or which range between 0 and 100, if you consider centiles): in this fashion it is more natural to aggregate and compare (you avoid any issue related to the variable binning).

7. Lines 120-126: you need to state here that the larger is the quantile (or rank), the better the geographical correspondence between maximum observed displacement and max (or at least large) forecast displacement.

8. Lines 199-206: the expected histogram for a random process would be a flat histogram (each bin is equiprobable), and you could compare your histogram to a flat one as described in Wilks (2019). I do however suggest to simply describe visually the intuitive result: your histogram in figure 5 shows a mode for the highest rank, which shows some skill in detecting the location of the maximum displacement. Rephrase (and join) these paragraphs. The reference is:

- Wilks, D, 2019: Indices of Rank Histogram Flatness and Their Sampling Properties. *Mon Wea Rev*, **147**: 763-769. https://doi.org/10.1175/MWR-D-18-0369.1

**Minor (technical) revisions:**

1. Lines 3 (and line 5): I suggest replacing "expansion" with "decline" and replacing "advancing" with "retreat". This is because in the first few sentences the authors relate with climate change, so it sounds a bit counter-intuitive to talk about sea-ice expansion (after reading the article it is clear that the methodology applies to both spring melting and autumn freeze-up, but in the abstract and introduction -if you maintain the climate flavour- you might prefer focussing on "decline")

2. Lines 28-32: citing literature from the verification community is welcome! You could add to Ebert and McBride (2000) also Davis et al (2006a,b), since MODE is now the most commonly used object-oriented verification method in the weather community. You could as well add that both these methods were designed for (several) precipitation-like (convex-shaped) features, and that there is no equivalent for a single linear feature such as ice-edge.

    a. Davis, C. A., B. G. Brown, and R. G. Bullock, 2006: Object-based verification of precipitation forecasts, Part I: Methodology and application to mesoscale rain areas. *Mon. Wea. Rev.* **134**, 1772 - 1784, doi: 10.1175/MWR3145.1.

    b. Davis, C. A., B. G. Brown, and R. G. Bullock, 2006: Object-based verification of precipitation forecasts, Part II: Application to convective rain systems. *Mon. Wea. Rev.* **134**, 1785 - 1795, doi: 10.1175/MWR3146.1

3. End of line 33: please be specific in stating that "We begin this study by presenting a new algorithm for assessing the quality of representation of *the displacement over time* of the sea-ice edge ... "
4. Line 34-36: please rephrase these two sentences to align with the suggestion of major revision 4b.
5. Line 39: replace "some idealized distributions are ..." with "an idealized ice-edge is"
6. Line 49 (and 70): I am sure the Hausdorff distance was introduced earlier than Dukhovskoy et al (2015), can you please provide the original reference?
7. Lines 78-83: you can join these two sentences in a single paragraph. You should replace "when time serie results are examined" with "when results aggregated over multiple cases over an extended time period", or something similar (the point is that you aggregate multiple cases, and not that you consider a time serie)
8. Lines 85-87: rephrase these two sentences to twin them with the beginning of Section 2.1, something like "In the previous section we focussed on measures which describe the displacement of a single (forecast or observed) ice-edge. In this section we extend these to assess the differences in the displacements of the forecast versus observed ice-edge."
9. Rephrase lines 88-89.
10. In the caption of Figure 2, lines 3-4, replace "the mean separation distance difference" with "A measure of the overall displacement difference" (be careful, that grey-shaded integral is not the mean displacement difference). The idea of using the area between the two curves (the grey shaded area) is excellent!
11. Line 92: replace "property" with "attribute" (verification term).
12. Line 93: I suggest writing "a simple measure which provide this type of information is ..."
13. Line 99: replace "site-specific" with "local".
14. Lines 99-100: write "... of the model ice edge in proximity of the maximum displacement found in the observations".
15. Line 110: "In order *to* examine ... "
16. The original reference for the rank (Talagrand) histogram is Talagrand et al (1999) and Anderson (1996)
    a. Talagrand, O., R. Vautard, and B. Strauss, 1999: Evaluation of probabilistic prediction systems. *Proc. Workshop on Predict- ability,* Reading, United Kingdom, ECMWF, 1–25.
    b. Anderson, J. L., 1996: A method for producing and evaluating probabilistic forecasts from ensemble model integrations. *J. Climate,* **9,** 1518–1530.
17. Line 171: change "integrated" to "interpolated" or "upscaled" (I imagine it is a mass-conservative up-scaling from the ice charts at 1km resolution to the SVIM 4km resolution domain). Is the smoothing (the second order checkboard suppression mentioned at line 161) performed on the observations too?
18. Line 172: what is meant here with "dry"?
19. Line 180: Eliminate "Category" and write "The distribution frequencies in Table 2 change only moderately when including open ocean boundaries and coastal lines."
20. Line 185: replace "unreasonable results" with "mis-matches ice-edge points".

21. Rephrase lines 188-189: I suppose you consider a fix number of nine ranks (and not nine randomly chosen points of the ice-edge).
22. Lines 193-198: please consider using quantiles, rather than ranks, as suggested in major comment 6.
23. The first paragraph of the conclusion need polishing / rephrasing.

---

## Author Comment (AC1)

**Author's response to Referee Comment 1**

I thank the referee for taking the time to review the manuscript, and providing a few comments on its contents.

   The reservations that the referee have, are regarding clarity. In this respect, two points are made. Should the referee have additional specifications about precisely what is unclear, I will do my best to respond to that as well.

1. The referee states that it is
   *not very evident how useful and applicable the method would be in practice*
   Section 3 (**A case study**) was written specifically with this topic in mind. Examples of the applicability of the method are given e.g. in Table 2, which provide statistics for edge displacements that allows an assessment of the general quality of such displacement from model results, and Figure 5 which contain information of the quality of the model's ability to reproduce the location of large ice edge displacements. Hence, the manuscript will not be modified in response to this item.

2. The referee states that it is
   *not clear what advantages or trade-offs the method can have over pre-existing verification metrics*
   In the event that the referee is aware of other methods that specifically target the quality of results for ice edge displacements from an expanding ice cover, I will evaluate the specific relevance of methods presented here. The situation is presently that I'm not aware of such a method. So, in response to this item, the following sentences will be introduced in the revision of the manuscript, in Section 4:
   *The results from these methods expand on existing validation metrics such as e.g. the Integrated Ice-Edge Error (Goessling et al., 2016) and the various ice edge metrics considered by Melsom et al. (2019): The present methods provide summary statistics for the quality of model results for ice edge displacements in the presence of an expanding sea ice cover, as exemplified by Table 2 and Fig. 5, that are not provided with existing metrics. This is of high relevance for planned or ongoing site specific activities in regions which can potentially become ice infested.*

---

## Author Comment (AC2)

**Author's response to Referee Comment 2**

I am grateful that the referee has taken the time to write a second review of the manuscript, providing helpful suggestions for making improvements to the manuscript. Below are my responses, which I hope will give answers to the referee's satisfaction. A number of modifications, mostly additions, have been included in the manuscript revision which is presently being prepared for submission.

1. From referee item 1:
   *[Section 3] contains a lot of quantitative information, and it may be helpful to know how the end-user might translate the information, for example the distribution in table 2.*
   I realize that the discussion on the results in Table 2 might benefit from additiopnal considerations. So in response to the referee's suggestion, a new paragraph has been added, with the following contents:
   *The conclusions that can be drawn from these results are that the largest expansion of sea ice extent in the model (SVIM) result are underestimations when compared with observations (ice chart data). This is the case for a general forecast bulletin, as the SVIM median is in the range 20 − 30 km, while the median of the ice chart data is in the range 30 − 40 km. The underestimation is also seen for extreme cases, as the frequency of maximum expansion exceeding 60 km is about five times as high for the ice chart data. Note that the ice chart data deviates from microwave products, particularly in the final months of the melting season (e.g. Sect. 6 in Melsom et al., 2019). Hence, the true sea ice extent is unknown.*

2. Also from referee item 1:
   *It would also be interesting to see if the ranks achieved by the model have a temporal or spatial pattern.*
   I agree that this is interesting, but it is also challenging, particularly with a very dynamic ice edge in a limited domain as in the 2 year case study in Section 3. (The fact that the domain is limited has implications for the degrees of freedom, given the spatial decorrelation length, as e.g. discussed in relation to equations 9 and 10.) In response to the reviewer's valid request, I have chosen to add results by splitting the domain in two parts, separated by the 40°E meridian. The results will be discussed in the revision, and presented in a new panel in Figure 5. I provide the revised version of Figure 5 at the end of this reply. The meridian of separation will be highlighted on the map in Figure 3.

3. From referee item 3:
   *[line 203] seems to imply that a random distribution of 235 integers between 0 and 9 will have 4.015 and 4.985 as the 0.005th and 0.995th percentiles respectively. I tried to reproduce this with various integer distributions with mean at 4.5 and cut-off at 0 and 9, but was not able to do so.*
   First, I realize that my reference to percentiles was incorrect, this will be rewritten to "*0.5$^{th}$ and 99.5$^{th}$ percentiles*". Secondly, the number of digits was unnecessary large, and the range will be rewritten to "*percentiles of ranks are 4.02 and 4.98, respectively*". Finally, regarding how these values can be found, first observe that for ranks, the underlying distribution is flat. Then, code in R for estimating the percentiles is as follows:

```
minVal <-    0
maxVal <-    9
nVal   <- 235
nCases <- 1000000
aveVal <- vector(mode="numeric",length=nCases)
for (n in 1:nCases) {
    values    <- sample(minVal:maxVal,nVal,replace=TRUE)
    aveVal[n] <- sum(values)/nVal
}
lim99lo <- round(0.005*nCases)
v99lo   <- sortVal[lim99lo]
lim99hi <- round(0.995*nCases)
v99hi   <- sortVal[lim99hi]
print(paste(v99lo,v99hi))
```

```
quit("no")
```

The final print statement should give values that can be rounded to 4.02 and 4.98, respectively.

- - - - - - - - - - - - -
**Revised version of Figure 5**

---

## Author Comment (AC3)

**Author's response to Referee Comment 3**

I sincerely appreciate the detailed comments and suggestions provided by the reviewer as Referee Comment 3. This input helped me in rewriting the manuscript to make it more accessible for readers of The Cryosphere and others. In my personal view, thanks to the efforts of the reviewer, the manuscript has now improved significantly from the initial submission. My detailed responses to the list of review items are provided below, with citations from the present referee comment in italics. In response to some of the concerns raised by the reviewer, the revised manuscript will include two appendices: Appendix A and B are attached at the end of this document.

**Major revision items**

1. **Section 2.1 needs major rewriting.** *Since the author aims to maximize generalization, the description is too generic and therefore it is not clear (until you read section 3). A not exhaustive list of suggestions (for improvement) are:*

   - *State more explicitly and since the beginning (e.g. at lines 33, in the paragraph at lines 39-45, and then at line 47) that you aim to verify the* displacement over time of the sea-ice edge, and compare the forecast versus observed displacements. *(e.g. at line 47 the phrasing "the difference in maximum edge displacement between two products" can be mis-interpreted as the "distance between observed and forecasted ice-edges").*

     The reviewer is correcting when pointing out that this aspect, which is central to the present work, was not given proper attention in the original manuscript. In the present revision this aspect is now described explicitly near the end of the Introduction section, and in the first paragraph of the following section.

   - *Do not use "product". You can use "dataset", "ice-edge" or other terms. You can as well refer to model (or forecast) and observation, explicitly.*

     I agree that the reviewer's suggestion is a better choice of words, and I have replaced 'product(s)' nearly everywhere in the manuscript, mostly by substitutions with 'data set(s)'.

   - *The titles of Sections 2.1 and 2.2 should include "displacement" somewhere, e.g. "2.1 verification measures of the displacement of a single ice-edge" and "2.2. comparison of the displacements of forecast and observed ice-edges"*

     I have rephrased the titles in question along the lines suggested by the reviewer. I have however used slightly different formulations.

   - *Lines 58-59: "$L^{(1)}, L^{(2)}$ denote the sea ice edges for two representations..." is not clear at all: here* **you need to state explicitly here that (1) indicate the time $t_0$ and (2) indicate the time $t_0+\triangle t$, and that you measure the ice-edge displacement that occurred between the times $t_0$ and $t_0+\triangle t$** *(for either the forecast or the observation ice-edge). In fact, I suggest to change all the notation here (otherwise the reader will naturally associate 1 for forecast and 2 to observations), replacing (1) with (t0) and (2) with (t0+$\triangle$t), and then $d_n^{2:1}$ can become $d_n^{\triangle t}$.*

     The description on lines 58-70 was meant as a general approach for computing distances between any two lines, e.g., edge lines in two different data sets or edge lines at different times within the same dataset. Nevertheless, I realize that this can potentially give rise to some confusion. Consequently, I have decided to follow the reviewer's suggestion, and later add a comment that the approach is not only valid for distances between edge lines at two different times, but also generally, for any set of two edge lines.

   - *The mathematical equations (3) and (4) need to be defined in a more mathematical rigorous way. Especially equation (4), the sign is not accounted for in the mathematical formula and there is a*

*missing absolute value $|.|$.*

**Regarding Eq. (3)**, I agree that there was a room for improvement. Consequently, I have rewritten the contents on l. 55-63 ("Denoting [...] distance of $z$.") to

Denoting the N grid cells that satisfy this condition for time $t$ by $e_1^{(t)}, e_2^{(t)}, \ldots, e_N^{(t)}$ the ice edge for time $t$ is then the line

$$L(t) = \{e_1^{(t)}, e_2^{(t)}, \ldots, e_N^{(t)}\}$$

This follows the algorithm presented in Melsom et al. (2019). Let $L(t_0), L(t_0 + \Delta t)$ denote the sea ice edges at times $t_0$ and $t_0 + \Delta t$, respectively. Furthermore, let $d_n^{\Delta t}$ be the displacement distance between a grid cell $e_n^{(t_0+\Delta t)}$ in $L(t_0 + \Delta t)$ and line $L(t_0)$, *i.e.*,

$$d_n^{\Delta t} = s_n \min ||e_n^{(t_0+\Delta t)} - L(t_0)||$$

where $s_n$ is +1 or -1 when $e_n^{(t_0+\Delta t)}$ is on the no ice or ice side of $L(t_0)$, respectively, explicitly defined by Eq. (4) below. $||z||$ is the Euclidean distance of $z$, and the length of dashed lines in Fig. 1 correspond to the displacements $\min ||e_n^{(t_0+\Delta t)} - L(t_0)||$ for selected cells $e_n^{(t_0+\Delta t)}$.

If we denote the sea ice concentration at the time $t_0$ for a grid cell $e_n^{(t_0+\Delta t)}$ (belonging to the ice edge at $t_0 + \Delta t$) by $c[e_n^{(t_0+\Delta t)}](t_0)$, $s_n$ is given as follows:

$$s_n = \begin{cases} -1 & \text{if } c[e_n^{(t_0+\Delta t)}](t_0) \geq c_{edge} \\ +1 & \text{if } c[e_n^{(t_0+\Delta t)}](t_0) < c_{edge} \end{cases}$$

**Regarding Eq. (4)**, I realize that the description of the quantity could lead to confusion, and I have rewritten (l. 63) "maximum distance" as "maximum expansion displacement". Eq. (4) is correct, as it returns the intended value. If $d_n = 1, 3, -5, d_{max} = 3$, and if $d_n = -1, -3, -5, d_{max} = -1$. This was by design, and was detailed on l. 65-67. The rationale for this definition was stated on l. 66-67 ("The definition of $s$ was designed so that $d_{max}^{2:1}$ will represent the displacement of the largest sea ice advance from $L^{(1)}$ to $L^{(2)}$.") The paragraph has been slightly rephrased in the revision, to make this even more clear.

- *Lines 62-63: the text is not precise; you need to state that in Eq. (3) you consider the minimum of all distances (between the single point in the edge at time $t_0 + \Delta t$ and all points of the edge at time $t_0$). Similarly, for Eq(4) you need to state that you consider the max (over all n) of the $d_n^{\Delta t}$*

  In the revision I have rewritten these items as suggested by the reviewer.

- *Lines 65: I believe this should state " ... will return the largest absolute positive value ..."*
  **AND**

- *Line 66: what happens if the $d_n^{2:1}$ are partially positive and partially negatives? Mention the canceling errors.*

  As explained for the item "*The mathematical equations* ... **Regarding Eq. (4)**" above, the statement the reviewer's assumption regarding l. 65 is wrong. This should be clear from the revisions introduced in response to that item. Regarding the question with reference to l. 66, this was in fact described on l. 66 ["Eq. (4) will return the largest positive value among $d_n^{2:1}$", i.e., if there is a mix, the largest positive value is returned, which is simply spelling out the definition provided in Eq. (4)]. I fail to see any relevance of canceling errors in this context.

2. *Section 2.1, Lines 70-75: **Table 1 (as well as Table 2** in Section 3) lists the frequencies of the histograms of the set of distances $d_n^{2:1}$ defned by Eq. (3), for the idealized model and observation ice-edges of figure 1. This is not clear from the text at line 75, nor from the caption (which I believe has a mistake, since it should refer to Eq. 3, and not Eq. 4) and the heading of the table is wrong (it should state "Frequencies" rather than "fraction of grid cells"). Here are my suggestions:*

- ***Representing a distribution with a table is quite unconventional: I strongly recommend plotting the histograms of the distances $d_n^{2:1}$ instead.*** *(You can choose if to leave the table or eliminate it, but please add the histogram). Please show the histograms also for Table 2.*

  These results are now shown as two histograms, which are displayed as Fig.s 2 and 5 in the revised manuscript.

- *Please rephrase the caption ("category distribution of displacement distances" is not clear; this can simply be "distribution of ice-edge displacements").*

  The caption has been rewritten as suggested by the reviewer.

- *Please rephrase lines 71-74: you can shrink it all as a couple of sentences such "The maximum distance in Eq. (4) provides a single measure to examine the ice-edge displacement. However, it can be more informative to analyse the whole distribution of the displacements $d_n^{2:1}$ defined by Eq. (3), rather than their maximum only. This can be done by analysing the histogram of the displacements (Figure HIST) and its corresponding frequencies (Table 1); the distribution of the displacements $d_n^{2:1}$ can as well be represented by their cumulative probability distribution (Figure 2)."*

  I agree that the reviewer's suggestion is a better choice of words, and have chosen to adopt it (with very slight modifications) in the revised manuscript.

- *Please rephrase line 75: "the distribution of selected distance categories" is not clear.*

  I have rewritten the entire paragraph in question (i.e., l. 71-77), and this phrase is no longer included in the manuscript.

3. *The **decorrelation length**, used to sub-sample the ice-edge, is lightly mentioned at lines 78-80. It is thereafter used (e.g. caption of Figure 2, and then more heavily at lines 110-115), however a more thorough definition and how the author calculate $\Delta n$ is missing in the article. Please add some text about it (maybe this can be done in an appendix).*

An appendix (A) with the mathematical formulation for computing the decorrelation length is included in the revised manuscript, as suggested by the reviewer.

4. *Section 2.3 is too long, needs rewording, and need to be accompanied by a visual example. I strongly suggest to:*

   a *summarize it in few sentences, such as in Melsom 2019, page 617, left column third paragraph ("A variant ..."). In the re-wording it is important that you still retain the explanation at lines 138-139, where you correctly state that you expect the distances to be smaller when adding ocean open boundaries and coastal lines (because when adding these artificial "fixed" edges you automatically include in your verification some perfectly matched edges, aka trivial skill).*

   b ***Include subsection 2.3 in Section 3**, after the data description and prior the results (aka after line 173 and before line 174). You can actually split section 3 in three subsections: 3a Sea-ice data description, 3b open boundaries and coasts, 3c verification results. In this fashion the reader has an immediate visual example (figure 3) on your need of adding these fictitious boundaries (especially when considering an Arctic sub-region).*

   c *The example presented at lines 180-187 and illustrated in figure 4 is excellent!*

These suggestions for changing the structure of the manuscript are well explained. Consequently, the revised manuscript is modified in response to items a and b, albeit somewhat differently from the reviewer's recommendations. I have decided to move the original subsection 2.3 to a new Appendix (B), where I also include a conceptual figure in order to illustrate the modifications to the algorithm in Section 2. I have also split Section 3 into two parts (not three, since the modifications due to open boundaries and coastlines are now extensively detailed in Appendix B).

5. *Give a new title to Section 3: Application of the new verification method on sea-ice forecast (this is not a "case study" ... also at line 211).*

   The title has been rewritten, using a slightly shorter phrase than proposed by the reviewer: "Application of the new validation method"

6. *Lines 115-119: **rather than considering the ranks**, and a variable numbers of bins, I suggest using the **quantiles**, which have a fix range in [0,1] (or which range between 0 and 100, if you consider centiles): in this fashion it is more natural to aggregate and compare (you avoid any issue related to the variable binning).*

   The use of quantiles makes sense when the degrees of freedom allows a large number of statistically independent values to be included in the analysis. In the present analysis, this is related to the decorrelation length $\Delta n$ along the ice edges. For the results in Section 3, the number of independent displacement distances is variable, and not large. This was why a rank histogram analysis using 10 bins was applied. An explanation along these lines have been added in the revised manuscript. The issue is further highlighted in a new analysis, when the domain is split in two (in response to a suggestion from another reviewer). In that case, the number of bins needed to be limited to eight in order to keep the majority of the dates in the analysis.

   Consequently, I have kept the approach of using rank histograms in the manuscript.

7. *Lines 120-126: you need to state here that the larger is the quantile (or rank), the better the geographical correspondence between maximum observed displacement and max (or at least large) forecast displacement.*

   I agree, in the revised manuscript I have included two sentences where these general properties are explained.

8. *Lines 199-206: the expected histogram for a random process would be a flat histogram (each bin is equiprobable), and you could compare your histogram to a flat one as described in Wilks (2019). I do however suggest to simply describe visually the intuitive result: your histogram in figure 5 shows a mode for the highest rank, which shows some skill in detecting the location of the maximum displacement. Rephrase (and join) these paragraphs. The reference is:*

   - *Wilks, D, 2019: Indices of Rank Histogram Flatness and Their Sampling Properties. Mon Wea Rev, 147: 763-769. https://doi.org/10.1175/MWR-D-18-0369.1*

   I have rewritten, and joined, the two paragraphs in question, trying to follow the reviewer's advice. The reference to equiprobable bins is included in Sect. 2.2. in the revised manuscript. A reference to Wilks (2019) has been included.

**Minor (technical) revisions**

1. *Lines 3 (and line 5): I suggest replacing "expansion" with "decline" and replacing "advancing" with "retreat". This is because in the first few sentences the authors relate with climate change, so it sounds a bit counter-intuitive to talk about sea-ice expansion (after reading the article it is clear that the methodology applies to both spring melting and autumn freeze-up, but in the abstract and introduction -if you maintain the climate flavour- you might prefer focussing on "decline")*

   This suggestion definitely warrant attention. The reference to climate change is made in response to the Arctic being expected to become a region with more activity over the coming decades, i.e. the phrase is a motivation for the topic at hand, but not to the present theory per se. The present work relates to evolutions over much shorter time scales, and the abstract of the present revision reflect this. The focus here is on the relevance for forecasted expansion of sea ice, which is a hazard for

open ocean operations in a polar environment. This was already reflected in the original manuscript, by the mathematical definitions in section 2.1, and in the text by e.g. pointing out that this leads to a *one-sided Hausdorff distance* variation, and also indicated by the choice of words elsewhere such as in the Abstract. Nevertheless, this reviewer comment, along with the question regarding Eq. (4) in Major item 1, made me realize that this aspect needs to be spelled out explicitly in Sect. 2. Consequently, I also rewrote the start of Sect. 2.1 to further highlight the focus on expansion.

2. *Lines 28-32: citing literature from the verification community is welcome! You could add to Ebert and McBride (2000) also Davis et al (2006a,b), since MODE is now the most commonly used object-oriented verification method in the weather community. You could as well add that both these methods were designed for (several) precipitation-like (convex-shaped) features, and that there is no equivalent for a single linear feature such as ice-edge.*

   a *Davis, C. A., B. G. Brown, and R. G. Bullock, 2006: Object-based verification of precipitation fore-casts, Part I: Methodology and application to mesoscale rain areas.* Mon. Wea. Rev. *textbf134, 1772 - 1784, doi: 10.1175/MWR3145.1.*

   b *Davis, C. A., B. G. Brown, and R. G. Bullock, 2006: Object-based verification of precipitation forecasts, Part II: Application to convective rain systems.* Mon. Wea. Rev. *textbf134, 1785 - 1795, doi: 10.1175/MWR3146.1*

   Thanks! In the revised manuscript a reference to Davis et al. (2006a) is added.

3. *End of line 33: please be specific in stating that "We begin this study by presenting a new algorithm for assessing the quality of representation of the displacement over time of the sea-ice edge ... "*

   This issue was attended to in relation to the reply to the first bullet point under Major item 1.

4. *Line 34-36: please rephrase these two sentences to align with the suggestion of major revision 4b.*

   The final paragraph in the Introduction is rewritten in the revised manuscript in order to reflect the changes in the manuscript's composition.

5. *Line 39: replace "some idealized distributions are ..." with "an idealized ice-edge is"*

   The phrase in question is rewritten in the revised manuscript, to "a set of idealized ice edges is".

6. *Line 49 (and 70): I am sure the Hausdorff distance was introduced earlier than Dukhovskoy et al (2015), can you please provide the original reference?*

   The original reference was not provided in any article where I've seen the Hausdorff distance applied (e.g. not in Dukhovskoy et al (2015), and not in either of the four articles that are cited on p. 5914 in that paper). According to Wikipedia (https://en.wikipedia.org/wiki/Grundz%C3%BCge_der_Mengenlehre), the original reference is
   Hausdorff, F.: Grundzüge der Mengenlehre, First edition, Verlag von Veit & Comp., Leipzig, Germany, 476pp, 1914
   This book is available from https://archive.org/details/grundzgedermen00hausuoft/page/n5/mode/2up
   This is a book of nearly 500 pages, and in German which is not my strongest side. After spending half an hour browsing the book half-blinded due to my limited skill in German, I was not able to find the definition. Consequently, I choose to refrain from listing this citation (but would be happy to include it, should the editor encourage me to include it). In any event, I find the description of the Hausdorff distance in Dukhovskoy et al (2015) useful, accessible, and easy to comprehend.

7. *Lines 78-83: you can join these two sentences in a single paragraph. You should replace "when time serie results are examined" with "when results aggregated over multiple cases over an extended time period", or something similar (the point is that you aggregate multiple cases, and not that you consider a time serie)*

The sentences have been rewritten, and put in a single paragraph, along the lines suggested by the reviewer.

8. *Lines 85-87: rephrase these two sentences to twin them with the beginning of Section 2.1, something like "In the previous section we focussed on measures which describe the displacement of a single (forecast or observed) ice-edge. In this section we extend these to assess the differences in the displacements of the forecast versus observed ice-edge."*

    I have rewritten the first paragraph of Sect. 2.2 along the lines suggested by the reviewer. However, the second sentence is kept (slightly rephrased), in order to link this paragraph explicitly to the set of ice edges displayed in Fig. 1.

9. *Rephrase lines 88-89.*

    These lines have been rephrased.

10. *In the caption of Figure 2, lines 3-4, replace "the mean separation distance difference" with "A measure of the overall displacement difference" (be careful, that grey-shaded integral is not the mean displacement difference). The idea of using the area between the two curves (the grey shaded area) is excellent!*

    For clarity, I have added "for the present subsample of ice edge grid cells" to make clear that I refer to the subsample, and not the full set. Aside from that, the integral indicated by the grey-shaded area is the mean displacement difference (when using the axis units in integration), as it is displayed for fractions (from 0 to 1): Say e.g. that the difference in displacements are 10 grids everywhere. Then, the area (the gray shaded integral) and the mean separation distance difference will both be 10 grid cells.

11. *Line 92: replace "property" with "attribute" (verification term).*

    Rewritten as suggested.

12. *Line 93: I suggest writing "a simple measure which provide this type of information is ..."*

    Rewritten along the lines suggested.

13. *Line 99: replace "site-specific" with "local".*

    Rewritten as suggested.

14. *Lines 99-100: write "... of the model ice edge in proximity of the maximum displacement found in the observations".*

    Rewritten as suggested.

15. *Line 110: "In order to examine ..."*

    Added 'to' as pointed out.

16. *The original reference for the rank (Talagrand) histogram is Talagrand et al (1999) and Anderson (1996)*

    a *Talagrand, O., R. Vautard, and B. Strauss, 1999: Evaluation of probabilistic prediction systems.* Proc. Workshop on Predictability, *Reading, United Kingdom, ECMWF, 125.*

    b *Anderson, J. L., 1996: A method for producing and evaluating probabilistic forecasts from ensemble model integrations.* J. Climate, **9**, 15181530

These references are added. Thanks!

17. *Line 171: change "integrated" to "interpolated" or "upscaled" (I imagine it is a mass-conservative up-scaling from the ice charts at 1km resolution to the SVIM 4km resolution domain). Is the smoothing (the second order checkboard suppression mentioned at line 161) performed on the observations too?*

    "Interpolation" is an estimation of an intermediate value based on a set of discrete existing values. I find this to be an improper description here, as grid cell values do not represent a point, but the average for a grid cell. I suppose "upscaled" is correct, but somewhat lacking in precision. At any rate, I find "integral" to be the proper designation here. For clarity, I rewrote the relevant passage to "results are integrated onto the SVIM domain using a mass conserving Riemann integral approach". The smoothing is not performed on the observations, since the data algorithm that produces the two-dimensional representation of sea ice concentration is not subject to the type of noise which arises from numeric dispersion from the model configuration. This is stated in the revised manuscript.

18. *Line 172: what is meant here with "dry"?*

    "Dry" has been rewritten to reflect that these are cells that lack proper values (due to the presence of land in the present context).

19. *Line 180: Eliminate "Category" and write "The distribution frequencies in Table 2 change only moderately when including open ocean boundaries and coastal lines."*

    Rewritten along the lines suggested; "Table 2" becomes "Fig. 5" and I have chosen to retain the reference to the algorithm, which now is detailed in Appendix B.

20. *Line 185: replace "unreasonable results" with "mis-matches ice-edge points"*

    "gives unreasonable results" has been rewritten as "mis-matches ice edge grid cells".

21. *Rephrase lines 188-189: I suppose you consider a fix number of nine ranks (and not nine randomly chosen points of the ice-edge).*

    The paragraph with these lines have been rephrased. This is where I set the rank size for the application example, as the reviewer points out (albeit to a rank size of ten, not nine). In the following text, the random draw required for the analysis is described, and an explanation for this particular choice of rank size is given.

22. *Lines 193-198: please consider using quantiles, rather than ranks, as suggested in major comment 6.*

    I disagree that the use of quantiles is a better approach, as explained in my response to major comment 6.

23. *The first paragraph of the conclusion need polishing / rephrasing.*

    The first paragraph has been rephrased. Note that in the revision the paragraph has also been rewritten in response to comments from another reviewer.

**Appendix A. Decorrelation length of displacements**

Assume that a we have a set of $N$ edge grid cells $e_n$ (i.e. satisfying Eq. (1)) that form a line

$$L = \{e_1, e_2, \ldots, e_N\} \tag{A1}$$

[revised manuscript text omitted]

---

## Author Response (AR1)

**Author's response for the handling Editor**

Dear Dr. Kaleschke,
Please find below my detailed responses to the three Referee Comments I have received. Citations from the referee comments are given in italics, and bold face page and line numbers in brackets refer to the places in the revised manuscript were the original manuscript was changed in response to the helpful comments and suggestions that I received. Individual replies to each referee comments have been posted in TC Discussions. This document lists these replies, with some modifications (e.g. longer quotes in the reply to RC1 and RC2 have been skipped below and replaced by references to the revised manuscript, and the inclusion of the Appendices in the reply to RC3 has been omitted.).

For the track-changes file, note that a few references in the original submission appear as double question marks in red. This is a feature of latexdiff which is due to restructuring of the manuscript. In the present case, Tables 1 and 2 have been removed (and replaced by Fig. 2 and 5), and subsection 2.3 has been removed (replaced by Appendix B). The pre-existing tables do not show up in the track-changes file.
Best regards,
Arne Melsom

**Author's response to Referee Comment 1**

1. The referee states that it is
   *not very evident how useful and applicable the method would be in practice*
   Section 3 (Application of the new validation method) was written specifically with this topic in mind. Examples of the applicability of the method are given e.g. in Fig. 5, which provide statistics for edge displacements that allows an assessment of the general quality of such displacement from model results, and Fig. 7 which contain information of the quality of the model's ability to reproduce the location of large ice edge displacements. Hence, the manuscript will not be modified in response to this item.

2. The referee states that it is
   *not clear what advantages or trade-offs the method can have over pre-existing verification metrics*
   In the event that the referee is aware of other methods that specifically target the quality of results for ice edge displacements from an expanding ice cover, I will evaluate the specific relevance of methods presented here. The situation is presently that I'm not aware of such a method. So, in response to this item, several new sentences are included in the first paragraph of Sect. 4. **[P12L234-238]**

**Author's response to Referee Comment 2**

1. From referee item 1:
   *[Section 3] contains a lot of quantitative information, and it may be helpful to know how the end-user might translate the information, for example the distribution in table 2.*
   I realize that the discussion on the results in Table 2 (Fig. 4 in the revised manuscript) might benefit from additiopnal considerations. So in response to the referee's suggestion, a new paragraph has been added. **[P9L187-P10L192]**

2. Also from referee item 1:
   *It would also be interesting to see if the ranks achieved by the model have a temporal or spatial pattern.*
   I agree that this is interesting, but it is also challenging, particularly with a very dynamic ice edge in a limited domain as in the 2 year case study in Section 3. (The fact that the domain is limited has implications for the degrees of freedom, given the spatial decorrelation length, as e.g. discussed in relation to equations 9 and 10.) In response to the reviewer's valid request, I have chosen to add results by splitting the domain in two parts, separated by the $40°$E meridian. The results are discussed

in the revision, and presented in a new panel in Fig. 7. The meridian of separation is highlighted on the map in Fig. 4. **[P12L222-229;FIG5,7]**

3. From referee item 3:

   *[line 203] seems to imply that a random distribution of 235 integers between 0 and 9 will have 4.015 and 4.985 as the 0.005th and 0.995th percentiles respectively. I tried to reproduce this with various integer distributions with mean at 4.5 and cut-off at 0 and 9, but was not able to do so.*

   First, I realize that my reference to percentiles was incorrect, this will be rewritten to "*0.5$^{th}$ and 99.5$^{th}$ percentiles*". Secondly, the number of digits was unnecessary large, and the range will be rewritten to "*percentiles of ranks are 4.02 and 4.98, respectively*". Finally, regarding how these values can be found, first observe that for ranks, the underlying distribution is flat. Then, code in R for estimating the percentiles is as follows:

```
minVal <-    0
maxVal <-    9
nVal   <- 235
nCases <- 1000000
aveVal <- vector(mode="numeric",length=nCases)
for (n in 1:nCases) {
    values    <- sample(minVal:maxVal,nVal,replace=TRUE)
    aveVal[n] <- sum(values)/nVal
}
lim99lo <- round(0.005*nCases)
v99lo   <- sortVal[lim99lo]
lim99hi <- round(0.995*nCases)
v99hi   <- sortVal[lim99hi]
print(paste(v99lo,v99hi))
quit("no")
```

The final print statement should give values that can be rounded to 4.02 and 4.98, respectively. **[P7L142,P11L218-219]**

**Author's response to Referee Comment 3**

**Major revision items**

1. ***Section 2.1 needs major rewriting.*** *Since the author aims to maximize generalization, the description is too generic and therefore it is not clear (until you read section 3). A not exhaustive list of suggestions (for improvement) are:*

   - *State more explicitly and since the beginning (e.g. at lines 33, in the paragraph at lines 39-45, and then at line 47) that you aim to verify the displacement over time of the sea-ice edge, and compare the forecast versus observed displacements. (e.g. at line 47 the phrasing "the difference in maximum edge displacement between two products" can be mis-interpreted as the "distance between observed and forecasted ice-edges").*

     The reviewer is correcting when pointing out that this aspect, which is central to the present work, was not given proper attention in the original manuscript. In the present revision this aspect is now described explicitly near the end of the Introduction section, and in the first paragraph of the following section. **[P2L36-37;46-48]**

   - *Do not use "product". You can use "dataset", "ice-edge" or other terms. You can as well refer to model (or forecast) and observation, explicitly.*

     I agree that the reviewer's suggestion is a better choice of words, and I have replaced 'product(s)' nearly everywhere in the manuscript, mostly by substitutions with 'data set(s)'. **[all sections]**

- *The titles of Sections 2.1 and 2.2 should include "displacement" somewhere, e.g. "2.1 verification measures of the displacement of a single ice-edge" and "2.2. comparison of the displacements of forecast and observed ice-edges"*

  I have rephrased the titles in question along the lines suggested by the reviewer. I have however used slightly different formulations. **[P2L49,P5L100]**

- *Lines 58-59: "$L^{(1)}$,$L^{(2)}$ denote the sea ice edges for two representations..." is not clear at all: here **you need to state explicitly here that (1) indicate the time $t_0$ and (2) indicate the time $t_0 + \triangle t$, and that you measure the ice-edge displacement that occurred between the times $t_0$ and $t_0 + \triangle t$** (for either the forecast or the observation ice-edge). In fact, I suggest to change all the notation here (otherwise the reader will naturally associate 1 for forecast and 2 to observations), replacing (1) with (t0) and (2) with (t0+$\triangle$t), and then $d_n^{2:1}$ can become $d_n^{\triangle t}$.*

  The description on lines 58-70 was meant as a general approach for computing distances between any two lines, e.g., edge lines in two different data sets or edge lines at different times within the same dataset. Nevertheless, I realize that this can potentially give rise to some confusion. Consequently, I have decided to follow the reviewer's suggestion, and later add a comment that the approach is not only valid for distances between edge lines at two different times, but also generally, for any set of two edge lines. **[EQ(2)-(11), EQ(B1)-(B8) notations & quantities in text; P4L98-99]**

- *The mathematical equations (3) and (4) need to be defined in a more mathematical rigorous way. Especially equation (4), the sign is not accounted for in the mathematical formula and there is a missing absolute value $|.|$.*

  **Regarding Eq. (3)**, I agree that there was a room for improvement. Consequently, I have rewritten the contents on l. 55-63 in the revised manuscript. **[P3L59-P4L72]**
  **Regarding Eq. (4)**, I realize that the description of the quantity could lead to confusion, and I have rewritten (l. 63) "maximum distance" as "maximum expansion displacement". Eq. (4) is correct, as it returns the intended value. If $d_n = 1, 3, -5, d_{max} = 3$, and if $d_n = -1, -3, -5, d_{max} = -1$. This was by design, and was detailed on l. 65-67. The rationale for this definition was stated on l. 66-67 ("The definition of $s$ was designed so that $d_{max}^{2:1}$ will represent the displacement of the largest sea ice advance from $L^{(1)}$ to $L^{(2)}$.") The paragraph has been slightly rephrased in the revision, to make this even more clear. **[P4L73,76]**

- *Lines 62-63: the text is not precise; you need to state that in Eq. (3) you consider the minimum of all distances (between the single point in the edge at time $t_0 + \Delta t$ and all points of the edge at time $t_0$). Similarly, for Eq(4) you need to state that you consider the max (over all n) of the $d_n^{\Delta t}$*

  In the revision I have rewritten these items as suggested by the reviewer. **[P3L66-67,P4L74]**

- *Lines 65: I believe this should state " ... will return the largest absolute positive value ..."*
  **AND**

- *Line 66: what happens if the $d_n^{2:1}$ are partially positive and partially negatives? Mention the canceling errors.*

  As explained for the item "*The mathematical equations ... **Regarding Eq. (4)**" above, the statement the reviewer's assumption regarding l. 65 is wrong. This should be clear from the revisions introduced in response to that item. Regarding the question with reference to l. 66, this was in fact described on l. 66 ["Eq. (4) will return the largest positive value among $d_n^{2:1}$", i.e., if there is a mix, the largest positive value is returned, which is simply spelling out the definition provided in Eq. (4)]. I fail to see any relevance of canceling errors in this context.

2. *Section 2.1, Lines 70-75: **Table 1 (as well as Table 2** in Section 3) lists the frequencies of the histograms of the set of distances $d_n^{2:1}$ defned by Eq. (3), for the idealized model and observation ice-edges of figure 1. This is not clear from the text at line 75, nor from the caption (which I believe*

*has a mistake, since it should refer to Eq. 3, and not Eq. 4) and the heading of the table is wrong (it should state "Frequencies" rather than "fraction of grid cells"). Here are my suggestions:*

- **Representing a distribution with a table is quite unconventional: I strongly recommend plotting the histograms of the distances** $d_n^{2:1}$ **instead.** *(You can choose if to leave the table or eliminate it, but please add the histogram). Please show the histograms also for Table 2.*

  These results are now shown as two histograms, which are displayed as Fig.s 2 and 5 in the revised manuscript. **[FIG2,5]**

- *Please rephrase the caption ("category distribution of displacement distances" is not clear; this can simply be "distribution of ice-edge displacements").*

  The caption has been rewritten as suggested by the reviewer. **[FIG2caption]**

- *Please rephrase lines 71-74: you can shrink it all as a couple of sentences such "The maximum distance in Eq. (4) provides a single measure to examine the ice-edge displacement. However, it can be more informative to analyse the whole distribution of the displacements* $d_n^{2:1}$ *defined by Eq. (3), rather than their maximum only. This can be done by analysing the histogram of the displacements (Figure HIST) and its corresponding frequencies (Table 1); the distribution of the displacements* $d_n^{2:1}$ *can as well be represented by their cumulative probability distribution (Figure 2)."*

  I agree that the reviewer's suggestion is a better choice of words, and have chosen to adopt it (with very slight modifications) in the revised manuscript. **[P4L85-88]**

- *Please rephrase line 75: "the distribution of selected distance categories" is not clear.*

  I have rewritten the entire paragraph in question (i.e., l. 71-77), and this phrase is no longer included in the manuscript. **[P4L82-84]**

3. *The* **decorrelation length***, used to sub-sample the ice-edge, is lightly mentioned at lines 78-80. It is thereafter used (e.g. caption of Figure 2, and then more heavily at lines 110-115), however a more thorough definition and how the author calculate* $\Delta n$ *is missing in the article. Please add some text about it (maybe this can be done in an appendix).*

   An appendix (A) with the mathematical formulation for computing the decorrelation length is included in the revised manuscript, as suggested by the reviewer. **[P13L265-276]**

4. *Section 2.3 is too long, needs rewording, and need to be accompanied by a visual example. I strongly suggest to:*

   a *summarize it in few sentences, such as in Melsom 2019, page 617, left column third paragraph ("A variant ..."). In the re-wording it is important that you still retain the explanation at lines 138-139, where you correctly state that you expect the distances to be smaller when adding ocean open boundaries and coastal lines (because when adding these artificial "fixed" edges you automatically include in your verification some perfectly matched edges, aka trivial skill).*

   b *__Include subsection 2.3 in Section 3__, after the data description and prior the results (aka after line 173 and before line 174). You can actually split section 3 in three subsections: 3a Sea-ice data description, 3b open boundaries and coasts, 3c verification results. In this fashion the reader has an immediate visual example (figure 3) on your need of adding these fictitious boundaries (especially when considering an Arctic sub-region).*

   c *The example presented at lines 180-187 and illustrated in figure 4 is excellent!*

   These suggestions for changing the structure of the manuscript are well explained. Consequently, the revised manuscript is modified in response to items a and b, albeit somewhat differently from the reviewer's recommendations. I have decided to move the original subsection 2.3 to a new Appendix

(B), where I also include a conceptual figure in order to illustrate the modifications to the algorithm in Section 2. I have also split Section 3 into two parts (not three, since the modifications due to open boundaries and coastlines are now extensively detailed in Appendix B). **[SEC3structure,P13L277-311,FIGB1]**

5. *Give a new title to Section 3: Application of the new verification method on sea-ice forecast (this is not a "case study" ... also at line 211).*

   The title has been rewritten, using a slightly shorter phrase than proposed by the reviewer: "Application of the new validation method" **[P7L150]**

6. *Lines 115-119: **rather than considering the ranks**, and a variable numbers of bins, I suggest using the **quantiles**, which have a fix range in [0,1] (or which range between 0 and 100, if you consider centiles): in this fashion it is more natural to aggregate and compare (you avoid any issue related to the variable binning).*

   The use of quantiles makes sense when the degrees of freedom allows a large number of statistically independent values to be included in the analysis. In the present analysis, this is related to the decorrelation length $\Delta n$ along the ice edges. For the results in Section 3, the number of independent displacement distances is variable, and not large. This was why a rank histogram analysis using 10 bins was applied. An explanation along these lines have been added in the revised manuscript. The issue is further highlighted in a new analysis, when the domain is split in two (in response to a suggestion from another reviewer). In that case, the number of bins needed to be limited to eight in order to keep the majority of the dates in the analysis. **[P10L204-P11L208]**

   Consequently, I have kept the approach of using rank histograms in the manuscript.

7. *Lines 120-126: you need to state here that the larger is the quantile (or rank), the better the geographical correspondence between maximum observed displacement and max (or at least large) forecast displacement.*

   I agree, in the revised manuscript I have included two sentences where these general properties are explained. **[P7L140-142]**

8. *Lines 199-206: the expected histogram for a random process would be a flat histogram (each bin is equiprobable), and you could compare your histogram to a flat one as described in Wilks (2019). I do however suggest to simply describe visually the intuitive result: your histogram in figure 5 shows a mode for the highest rank, which shows some skill in detecting the location of the maximum displacement. Rephrase (and join) these paragraphs. The reference is:*

   - *Wilks, D, 2019: Indices of Rank Histogram Flatness and Their Sampling Properties. Mon Wea Rev, 147: 763-769. https://doi.org/10.1175/MWR-D-18-0369.1*

   I have rewritten, and joined, the two paragraphs in question, trying to follow the reviewer's advice. The reference to equiprobable bins is included in Sect. 2.2. in the revised manuscript. A reference to Wilks (2019) has been included. **[P11L216-221]**

**Minor (technical) revisions**

1. *Lines 3 (and line 5): I suggest replacing "expansion" with "decline" and replacing "advancing" with "retreat". This is because in the first few sentences the authors relate with climate change, so it sounds a bit counter-intuitive to talk about sea-ice expansion (after reading the article it is clear that the methodology applies to both spring melting and autumn freeze-up, but in the abstract and introduction -if you maintain the climate flavour- you might prefer focussing on "decline")*

   This suggestion definitely warrant attention. The reference to climate change is made in response to the Arctic being expected to become a region with more activity over the coming decades, i.e. the

phrase is a motivation for the topic at hand, but not to the present theory per se. The present work relates to evolutions over much shorter time scales, and the abstract of the present revision reflect this. The focus here is on the relevance for forecasted expansion of sea ice, which is a hazard for open ocean operations in a polar environment. This was already reflected in the original manuscript, by the mathematical definitions in section 2.1, and in the text by e.g. pointing out that this leads to a *one-sided Hausdorff distance* variation, and also indicated by the choice of words elsewhere such as in the Abstract. Nevertheless, this reviewer comment, along with the question regarding Eq. (4) in Major item 1, made me realize that this aspect needs to be spelled out explicitly in Sect. 2. Consequently, I also rewrote the start of Sect. 2.1 to further highlight the focus on expansion. **[P1L3-6,P2L50-53]**

2. *Lines 28-32: citing literature from the verification community is welcome! You could add to Ebert and McBride (2000) also Davis et al (2006a,b), since MODE is now the most commonly used object-oriented verification method in the weather community. You could as well add that both these methods were designed for (several) precipitation-like (convex-shaped) features, and that there is no equivalent for a single linear feature such as ice-edge.*

   a *Davis, C. A., B. G. Brown, and R. G. Bullock, 2006: Object-based verification of precipitation forecasts, Part I: Methodology and application to mesoscale rain areas.* Mon. Wea. Rev. *textbf134, 1772 - 1784, doi: 10.1175/MWR3145.1.*

   b *Davis, C. A., B. G. Brown, and R. G. Bullock, 2006: Object-based verification of precipitation forecasts, Part II: Application to convective rain systems.* Mon. Wea. Rev. *textbf134, 1785 - 1795, doi: 10.1175/MWR3146.1*

   Thanks! In the revised manuscript a reference to Davis et al. (2006a) is added. **[P2L31-32]**

3. *End of line 33: please be specific in stating that "We begin this study by presenting a new algorithm for assessing the quality of representation of the displacement over time of the sea-ice edge ... "*

   This issue was attended to in relation to the reply to the first bullet point under Major item 1.

4. *Line 34-36: please rephrase these two sentences to align with the suggestion of major revision 4b.*

   The final paragraph in the Introduction is rewritten in the revised manuscript in order to reflect the changes in the manuscript's composition. **[P2L37-39]**

5. *Line 39: replace "some idealized distributions are ..." with "an idealized ice-edge is"*

   The phrase in question is rewritten in the revised manuscript, to "a set of idealized ice edges is". **[P2L41]**

6. *Line 49 (and 70): I am sure the Hausdorff distance was introduced earlier than Dukhovskoy et al (2015), can you please provide the original reference?*

   The original reference was not provided in any article where I've seen the Hausdorff distance applied (e.g. not in Dukhovskoy et al (2015), and not in either of the four articles that are cited on p. 5914 in that paper). According to Wikipedia (`https://en.wikipedia.org/wiki/Grundz%C3%BCge_der_Mengenlehre`), the original reference is
   Hausdorff, F.: Grundzüge der Mengenlehre, First edition, Verlag von Veit & Comp., Leipzig, Germany, 476pp, 1914
   This book is available from `https://archive.org/details/grundzgedermen00hausuoft/page/n5/mode/2up`
   This is a book of nearly 500 pages, and in German which is not my strongest side. After spending half an hour browsing the book half-blinded due to my limited skill in German, I was not able to find the definition. Consequently, I choose to refrain from listing this citation (but would be happy to include it, should the editor encourage me to do so). In any event, I find the description of the Hausdorff distance in Dukhovskoy et al (2015) useful, accessible, and easy to comprehend.

7. *Lines 78-83: you can join these two sentences in a single paragraph. You should replace "when time serie results are examined" with "when results aggregated over multiple cases over an extended time period", or something similar (the point is that you aggregate multiple cases, and not that you consider a time serie)*

The sentences have been rewritten, and put in a single paragraph, along the lines suggested by the reviewer. **[P4L89-93]**

8. *Lines 85-87: rephrase these two sentences to twin them with the beginning of Section 2.1, something like "In the previous section we focussed on measures which describe the displacement of a single (forecast or observed) ice-edge. In this section we extend these to assess the differences in the displacements of the forecast versus observed ice-edge."*

I have rewritten the first paragraph of Sect. 2.2 along the lines suggested by the reviewer. However, the second sentence is kept (slightly rephrased), in order to link this paragraph explicitly to the set of ice edges displayed in Fig. 1. **[P5L101-103]**

9. *Rephrase lines 88-89.*

These lines have been rephrased. **[P5L108-109]**

10. *In the caption of Figure 2, lines 3-4, replace "the mean separation distance difference" with "A measure of the overall displacement difference" (be careful, that grey-shaded integral is not the mean displacement difference). The idea of using the area between the two curves (the grey shaded area) is excellent!*

For clarity, I have added "for the present subsample of ice edge grid cells" to make clear that I refer to the subsample, and not the full set. Aside from that, the integral indicated by the grey-shaded area is the mean displacement difference (when using the axis units in integration), as it is displayed for fractions (from 0 to 1): Say e.g. that the difference in displacements are 10 grids everywhere. Then, the area (the gray shaded integral) and the mean separation distance difference will both be 10 grid cells.

11. *Line 92: replace "property" with "attribute" (verification term).*

Rewritten as suggested. **[P5L112]**

12. *Line 93: I suggest writing "a simple measure which provide this type of information is ..."*

Rewritten along the lines suggested. **[P5L113]**

13. *Line 99: replace "site-specific" with "local".*

Rewritten as suggested. **[P5L119]**

14. *Lines 99-100: write "... of the model ice edge in proximity of the maximum displacement found in the observations".*

Rewritten as suggested. **[P5L119-120]**

15. *Line 110: "In order to examine ..."*

Added 'to' as pointed out. **[P6L131]**

16. *The original reference for the rank (Talagrand) histogram is Talagrand et al (1999) and Anderson (1996)*

a *Talagrand, O., R. Vautard, and B. Strauss, 1999: Evaluation of probabilistic prediction systems.* Proc. Workshop on Predictability, *Reading, United Kingdom, ECMWF, 125.*

b *Anderson, J. L., 1996: A method for producing and evaluating probabilistic forecasts from ensemble model integrations.* J. Climate, **9**, 15181530

These references are added. Thanks! **[P7L137-138]**

17. *Line 171: change "integrated" to "interpolated" or "upscaled" (I imagine it is a mass-conservative upscaling from the ice charts at 1km resolution to the SVIM 4km resolution domain). Is the smoothing (the second order checkboard suppression mentioned at line 161) performed on the observations too?*

    "Interpolation" is an estimation of an intermediate value based on a set of discrete existing values. I find this to be an improper description here, as grid cell values do not represent a point, but the average for a grid cell. I suppose "upscaled" is correct, but somewhat lacking in precision. At any rate, I find "integral" to be the proper designation here. For clarity, I rewrote the relevant passage to "results are integrated onto the SVIM domain using a mass conserving Riemann integral approach". The smoothing is not performed on the observations, since the data algorithm that produces the two-dimensional representation of sea ice concentration is not subject to the type of noise which arises from numeric dispersion from the model configuration. This is stated in the revised manuscript. **[P8L174,P8L163-164]**

18. *Line 172: what is meant here with "dry"?*

    "Dry" has been rewritten to reflect that these are cells that lack proper values (due to the presence of land in the present context). **[P8L174-175]**

19. *Line 180: Eliminate "Category" and write "The distribution frequencies in Table 2 change only moderately when including open ocean boundaries and coastal lines."*

    Rewritten along the lines suggested; "Table 2" becomes "Fig. 5" and I have chosen to retain the reference to the algorithm, which now is detailed in Appendix B. **[P10L193]**

20. *Line 185: replace "unreasonable results" with "mis-matches ice-edge points"*

    "gives unreasonable results" has been rewritten as "mis-matches ice edge grid cells". **[P10L198]**

21. *Rephrase lines 188-189: I suppose you consider a fix number of nine ranks (and not nine randomly chosen points of the ice-edge).*

    The paragraph with these lines have been rephrased. This is where I set the rank size for the application example, as the reviewer points out (albeit to a rank size of ten, not nine). In the following text, the random draw required for the analysis is described, and an explanation for this particular choice of rank size is given. **[P10L202-P11L208]**

22. *Lines 193-198: please consider using quantiles, rather than ranks, as suggested in major comment 6.*

    I disagree that the use of quantiles is a better approach, as explained in my response to major comment 6.

23. *The first paragraph of the conclusion need polishing / rephrasing.*

    The first paragraph has been rephrased. Note that in the revision the paragraph has also been rewritten in response to comments from another reviewer. **[P12L231-238]**

---

## Referee Report (RR1)

The study presents a new method for assessing forecasts of an expanding sea ice front by measuring the displacement of the edge. The method is promising and the author has done major revisions to the manuscript, thereby making the article suitable for publication in the Cryosphere. I address the response that the author has made to my comments and mention other minor points here:

1. In the revised version, the author has added more information regarding the applicability and the results derived from the method, thus issues with understanding the use scenario of the method have been addressed.

2. The author added lines to highlight how the method mentioned here is different from existing validation metrics. This issue has therefore been resolved.

3. Additional information has been added to the validation results in section 3.2 showing clearly the evaluation of the displacement distributions. The author has also replaced table 2 with Figure 5, which makes reading the results easier and intuitive.

4. In response to comment 2 (RC2), the author has included a subdivision of the validation domain into an eastern and western part, thereby including a spatial comparison of the results. While the average ranks of the model displacement vary only slightly between the two sections, the rank histogram in figurer 7 shows that some slots have a larger frequency difference. The subdivision is therefore a good addition to the analysis.

5. The author has been kind enough to clear my misunderstanding regarding the distribution described in line 219 and also detailed the process how they arrived at the numbers given. The updated percentiles are correct.

6. At various points in the manuscript, the author has referred to the methods described in the paper as "present methods". While this is generally clear, it can be confused with methods that are already present, in literature or common scientific practice. For example, in lines 235-237 of the new manuscript: "The present methods provide …. that are not provided with existing metrics" could be made clearer by instead using the term "The methods presented here provide…." or another suitable phrasing.

---

## Referee Report (RR2)

Revision CryoSphere-tc-2020-361
Author: A. Melson;     Title: Edge displacement scores

**Overall general comments:**

This study introduces a new distance-based approach for assessing the quality of the representation of the displacement over time of the sea-ice edge. The methodology introduced is innovative and scientifically sound. The manuscript is quite improved with respect to the first submissions. However some parts have become too long (or the information is fragmented across the article). I am therefore suggesting again major revisions (which are however less major than the first round of revisions, and less numerous too). Most of them aim to improve the exposition and article organization (e.g. shortening and moving the content of Appendix B in the article main body text). After these are performed, I'd be happy to recommend the manuscript for publication in the CryoSphere.

**Major Revisions:**

1. line 81, remove the reference to Dukhovskoy here (you already reference him at line 52, it is not necessary to repeat this), and instead add a sentence (at line 81) which explains the difference between your metric and the Hausdorff one, e.g. ", which guarantees symmetry in the distance when swapping the two compared datasets (usually an observed and a modelled feature). The distance measure introduced in this study, on the other hand, voluntary does not aim for symmetry, since it compares the position of the same feature at two different times, hence describing the feature displacement."

2. lines 85-88: please expand on the description and interpretation of Figure 2 and 3 for the idealized case study (either here or in the next section, at lines 104-107). I actually believe that **analysing the distribution of all distances is far more informative than analysing the maximum distance only**: this is why in my view **it is important you give more weight to this aspect of your technique**.

3. The authors moved (and expanded) the content of the original session 2.3 (in the March 2021 manuscript) to appendix B. Now the reading of the manuscript is much more difficult, because the information is fragmented everywhere. Moreover, I think the original session 2.3 (in the March 2021 manuscript) was better than the actual appendix B (which is overly-long). **I suggest putting back the material of appendix B in the text**, **but not as long as the appendix** (I actually already commented in March that the section 2.3 was too long). As commented for the section 2.3 of the March 2021 manuscript, I strongly suggest to:

   a) Insert a section between sections 3.1 and 3.2, where the material of the old section 2.3 will be described (since this is a technical part of the technique, I suggest describing it within the application part) and entitle it "Open ocean boundaries and coasts". Essentially you **split section 3 in three subsections: 3a Sea-ice data description, 3b open ocean boundaries and coasts, 3c verification results**. In this fashion the reader has also an immediate visual example (figure 4) on your need of adding these artificial ice-edges (especially when considering an Arctic sub-region).

b) summarize the content of this section in **few** sentences, such as in Melsom 2019, page 617, left column third paragraph ("A variant … "). Essentially all you have to say is that the ocean open boundaries as well as the coastal lines are added into the ice-edge definition. In the re-wording it is important that you explain that you expect the distances to be smaller when adding ocean open boundaries and coastal lines (because when adding these artificial "fixed" edges you automatically include in your verification some perfectly matched edges, aka trivial skill).

c) There are **too many equations** (in appendix B, but also in the past Section 2.3). **The only one you really need to retain is B7 (Equation 16 of the March 2021 manuscript)**, essentially you can explain in the text -simply verbally- that you add to the ice edge the costal line and ocean open boundary, and then you re-evaluate all the statistics as described in section 2.1 and 2.2.

d) The example presented at lines 193-200 (lines 180-187 in the March 2021 manuscript) and illustrated in figure 6 is excellent: I suggest you insert this example in this new section 3.2, since it illustrates why you need to include the ocean open boundaries and coasts. You might want to better phrase it, and explain why the ice edges are "mis-matched".

e) In conclusion, **the resulting new text should be shorter** than the current Appendix B (and possibly also shorter than the past section 2.3, in the March 2021 manuscript).

f) **If the authors decide to leave the Appendix B**, then they should not introduce the issue in section 2.1 (eliminate lines 94-97), but in the Section 3 (so they have the Figure 4 and 6 to refer to). As an example, after the sentence at lines 178-180 you should illustrate the example at lines 196-200 and then conclude stating "To address this issue, the ocean open boundaries as well as the coastal lines are added into the ice-edge definition (see Appendix B for details)". (You can merge into the text of lines 178-180 also some of the text currently at lines 94-97; also, remember to explain why the ice-edges are "mis-matched", at line 198).

g) I am pleased that you show in section 3 that **the results obtained adding the coastal lines and open ocean boundaries are similar to the ones you obtain without these "artificial" ice edge** extensions (e.g. lines 280-281 or 193-194): please make sure you keep stating this in the revised manuscript.

4. **I won't recommend using the average** for describing an histogram (or a distribution) which is not gaussian (or even bimodal, as for the obs). I believe that **the "mode" value (the one with the highest frequency),** or even the median, would be better indicators. Please change lines 185-186 accordingly (talk about the mode and eliminate the discussion about the average). similarly, at lines 216-221: retain the first two sentences as they are (I like you essentially compare the accumulated frequencies for ranks below and above the 50%ile). The subsequent lines (218-221) are not entirely clear: do you use the methods of Wilks (2019)? if so, please put the reference up front and rephrase the whole text, so that it becomes more clear. Similarly, at lines 226-229 you perform an identical analysis, but you are missing the conclusions (one sentence, stating that these findings show that the model well depicts the position of the max displacement).

**Minor revisions:**

5. Some suggested re-phrasing in the Introduction:
   a) line 18 and 19, rephrase "in their examination" (e.g. "to analyse the predictability of sea-ice edge")
   b) line 37-38, write "The method is described in Section 2 with an idealized case study."
   c) line 38, replace "in an examination of displacements" with "to analyse displacements … "
   d) line 39: write "Technical details … "
6. rephrase lines 47-48.
7. line 60: replace "line" with "curve" (possibly everywhere, since ice edge is rarely straight).
8. Suggested rephrasing for lines 67-74: line 67, end this sentence after "respectively"; then place Equation (4) followed by the text at line 70 as "where we denote the sea ice concentration … (t0)." Then you start a new paragraph introducing Figure 1: please expand a bit with respect to the text in line 68-69, and then concatenate with the text at lines 73-74. Example, start with "Figure 1 shows an idealized example where a modeled and an observed sea ice edge are displaced. The length of the dashed lines correspond to … . We then introduce the maximum expansion displacement as … ").
9. rephrase lines 91-93.
10. In light of major comment 3, eliminate lines 94-97.
11. I suggest eliminating also lines 98-99, and move this comment in the conclusions.
12. lines 108-111 repeat the same concepts stated in the previous paragraph (lines 104-107). Eliminate one of the two paragraphs, but (in light of Major comment 2) I would like you to expand and describe the result of Figures 2 and 3 more in detail.
13. please rephrase lines 145-146, e.g. "we randomly subsample a fixed number of intervals from Eq (11), so that the number bins is equal across different cases and results can be aggregated".
14. line 149, add "which reflects a forecast poor positioning of the maximum displacement"
15. eliminate lines 191-192 (it seems out of context here). Maybe this sentence is more suited in the data description, section 3.1?
16. Rephrase lines 202-203, e.g. "We consider a fix number of 10 bins for the present investigation, Hence … nine values are randomly selected from the displacements in Eq. (11)".
17. Lines 214-215 are not clear, rephrase (or eliminate) them.
18. Line 216: rephrase "in ranks 5-9 *than* ranks 0-4".
19. The first paragraph of the conclusions is weak, and can be improved.
20. In the text at lines 248-252 state explicitly that ocean open boundaries and coastal lines become part of the ice edge (rather than stating "a modification to the algorithm was introduced").
21. Figure 7: why when considering the whole domain there is a peak in the 4-5 ranks, whereas when considering the two separate domains this disappear? Are there still mis-matched ice edges?

---

## Referee Report (RR3)

Revision CryoSphere-tc-2020-361
Author: A. Melson;     Title: Edge displacement scores

**Overall general comments:**
This study introduces a new distance-based approach for assessing the quality of the representation of the displacement over time of the sea-ice edge. The methodology introduced is innovative and scientifically sound, the exposition is clear and the article well organized.
I suggest some very minor revisions, after which, I recommend the manuscript for publication in the CryoSphere.

**Very Minor suggestions:**

line 71: write " … for one product, and as illustrative example, we focus on the ice edge … "
line 98: replace "introduced" with "analysed".
line 99: write " … to compare model displacement distances with … "
line 102: write " … that span each … "
line 104: after describing the shift, add ", indicating an overprediction of the sea-ice displacement".
line 169: I sugest replacing the last sentence of this paragraph by a sentence stating that "Hence the sea-ice edge observations are estimated with a not-known uncertainty" (or similar).
Section 3.2 I thank you the author for introducing this section. I have some suggestions for rewording it:

"For some cases, the algorithm described in section 2.1 does not describe properly the true displacement. This particularly affects the maximum value as defined by Eq. (5). We illustrate this issue with a case study displayed in Figure 5, showing the 24-hour change in the ice-edge position from the 23$^{rd}$ to the 24$^{th}$ of October 2001. In this case the ice edge was displaced into the verification domain across an open boundary to the north. The general algorithm in Section 2.1 mis-matches the ice edge grid-cell (since cannot see the ice beyond the open boundary), and leads to an unrealistic maximum ice edge displacement of 285km, as given by the thick black line close to the sub-domain's north border.
In order to address this issue, a modification of the algorithm was implemented, in which ocean open boundaries are considered as "continuation" of the ice edge. The modified algorithm is described in full details in Appendix B. For the illustrated case study, the maximum ice edge displacement calculated with the modified algorithm becomes 79 km, and is indicated by the red line in the eastern part of the verification domain.
It must be noted that if the ice is advected into the domain, the distances associated with such a displacement will be underestimated, since the real position of the ice edge outside of the analysis domain at t0 is unknown. Other situations where unrealistic representations for displacements may also arise are when ice freezes along the coast, e.g. due to colder air in the vicinity of continents, or less salty water masses close to the coastline. This issue is treated analogously to advection across an open boundary, by considering coastlines as continuation of the ice edge (see Appendix B for details)."

line 208-209: replace "results are underestimations" with "are underestimated".
line 215: remove "Moreover"; line 216: replace "." with ":" and write "only 235 … ".
line 227: write " … indicate some skill for SVIM in detecting the location … ".
lines 243-244: write "Note that the distribution peak at ranks 4-5 for the full domain disappears when considering the rank frequencies for the two subdomains."
line 246: start the sentence with "Whereas, the ranges … "
line 247: replace "become" with "are".

I suggest re-wording lines 250-252 as follow: "The algorithm computes different attributes of the ice edge displacement, such as the maximum and the distribution of the distances. Then, different methodologies for comparing these attributes are introduced. The method introduced enables the assessment of the forecast quality for the displacement of the ice edge, and expands on existing validation metrics …"
lines 261-262: write " … perimeter of any physical variable or feature that can be represented by a spatially continuous binary field. Stratiform precipitation is an example of another physical variable for which the method presented here could be suitable."
line 267-268: write " … the original algorithm described in section 2.1 and 2.2 can sometimes mis-match ice cells, and hence diagnose unrealistic ice edge displacements, which may yield to mis-leading results."
line 269: write "open boundary" (eliminate "model domain" … you could add "ocean").
line 271-272: write " … as shown by the case study illustrated in Figure 5."

---

## Author Response (AR2)

**Author's response for the handling Editor**

Dear Dr. Kaleschke,
Please find below my detailed responses to the two Referee Reports I have received for the revised manuscript. Below, citations from the referee comments are given in italics, and bold face page and line numbers in brackets refer to the places in the revised manuscript were the original manuscript was changed in response to the helpful comments and suggestions that I received.

Note that the order of Figures 5 and 6 are swapped from the initial revision. This has an effect on the document where differences are displayed.

Best regards,
Arne Melsom

**Author's response to Referee Report 1**

I thank the referee for taking the time to review the revised manuscript, and for providing a list of six comments. Comments 1-5 don't ask for additional changes to the manuscript. Accordingly, this response only concerns the final item.

6. *At various points in the manuscript, the author has referred to the methods described in the paper as "present methods". While this is generally clear, it can be confused with methods that are already present, in literature or common scientific practice. For example, in lines 235-237 of the new manuscript: "The present methods provide .... that are not provided with existing metrics" could be made clearer by instead using the term "The methods presented here provide...." or another suitable phrasing.*

   I agree. The phrasing has been revised according to this suggestion, in three places in the updated revision. **[P13L254,P13L262,P13L273]**

**Author's response to Referee Report 2**

I am very grateful that the referee has taken the time to write a second review of the manuscript, providing helpful suggestions for making improvements to the manuscript. Below are my responses, which I hope will give answers to the referee's satisfaction. The modifications to the manuscript that the comments and suggestions have led to, are detailed below.

**Major Revisions**

1. *line 81, remove the reference to Dukhovskoy here (you already reference him at line 52, it is not necessary to repeat this), and instead add a sentence (at line 81) which explains the difference between your metric and the Hausdorff one, e.g. ", which guarantees symmetry in the distance when swapping the two compared datasets (usually an observed and a modelled feature). The distance measure introduced in this study, on the other hand, voluntary does not aim for symmetry, since it compares the position of the same feature at two different times, hence describing the feature displacement."*

   The reference to Dukhovskoy has been removed here, and an explanation following the reviewer's suggestions has been included. **[P4L81-83]**

2. *lines 85-88: please expand on the description and interpretation of Figure 2 and 3 for the idealized case study (either here or in the next section, at lines 104-107). I actually believe that **analysing***

*the distribution of all distances is far more informative than analysing the maximum distance only:* this is why in my view **it is important you give more weight to this aspect of your technique.**

I find that the description and interpretations that the reviewer suggest, is best placed in the latter of the two suggested places in the manuscript. Moreover, re-reading the two paragraphs on lines 104-111, I also realize that information was simply repeated in "neighboring sentences", which is not at all an optimal presentation. Next, I agree with the reviewer that the discussion here beyond the topic of maximum displacements was insufficient. In the present revision, topics such as the uneven distribution of distances, and the contrasts in frequencies of short displacements are discussed **[P5L99-109]**. Nevertheless, I find the aspect of the representation of maximum displacements to be the most informative from a user perspective: The ice edge is very dynamic, and in reality the historical data and time series with corresponding model results for the conditions in a specific geographical location at a given time of the year will be much too sparse to provide reliable statistics for the model performance. So from a user perspective, the "worst case" statistics of maximum displacements are likely the most valuable information at hand.

3. *The authors moved (and expanded) the content of the original session 2.3 (in the March 2021 manuscript) to appendix B. Now the reading of the manuscript is much more difficult, because the information is fragmented everywhere. Moreover, I think the original session 2.3 (in the March 2021 manuscript) was better than the actual appendix B (which is overly-long). I suggest putting back the material of appendix B in the text, but not as long as the appendix (I actually already commented in March that the section 2.3 was too long). As commented for the section 2.3 of the March 2021 manuscript, I strongly suggest to:*

    a *Insert a section between sections 3.1 and 3.2, where the material of the old section 2.3 will be described (since this is a technical part of the technique, I suggest describing it within the application part) and entitle it "Open ocean boundaries and coasts". Essentially you **split section 3 in three subsections: 3a Sea-ice data description,** **3b open ocean boundaries and coasts,** **3c verification results.** In this fashion the reader has also an immediate visual example (figure 4) on your need of adding these artificial ice-edges (especially when considering an Arctic sub-region).*

    b *summarize the content of this section in __few__ sentences, such as in Melsom 2019, page 617, left column third paragraph ("A variant ... "). Essentially all you have to say is that the ocean open boundaries as well as the coastal lines are added into the ice-edge definition. In the rewording it is important that you explain that you expect the distances to be smaller when adding ocean open boundaries and coastal lines (because when adding these artificial "fixed" edges you automatically include in your verification some perfectly matched edges, aka trivial skill).*

    c *There are **too many equations** (in appendix B, but also in the past Section 2.3). **The only one you really need to retain is B7 (Equation 16 of the March 2021 manuscript),** essentially you can explain in the text -simply verbally- that you add to the ice edge the coastal line and ocean open boundary, and then you re-evaluate all the statistics as described in section 2.1 and 2.2.*

    d *The example presented at lines 193-200 (lines 180-187 in the March 2021 manuscript) and illustrated in figure 6 is excellent: I suggest you insert this example in this new section 3.2, since it illustrates why you need to include the ocean open boundaries and coasts. You might want to better phrase it, and explain why the ice edges are "mis-matched".*

    e *In conclusion, **the resulting new text should be shorter** than the current Appendix B (and possibly also shorter than the past section 2.3, in the March 2021 manuscript).*

    f ***If the authors decide to leave the Appendix B,** then they should not introduce the issue in section 2.1 (eliminate lines 94-97), but in the Section 3 (so they have the Figure 4 and 6 to refer to). As an example, after the sentence at lines 178-180 you should illustrate the example at lines 196-200 and then conclude stating "To address this issue, the ocean open boundaries as well as the coastal lines are added into the ice-edge definition (see Appendix B for details)". (You can merge into the text of lines 178-180 also some of the text currently at lines 94-97; also, remember to explain why the ice-edges are "mis-matched", at line 198).*

g *I am pleased that you show in section 3 that **the results obtained adding the coastal lines and open ocean boundaries are similar to the ones you obtain without these "artificial" ice edge** extensions (e.g. lines 280-281 or 193-194): please make sure you keep stating this in the revised manuscript.*

Several of the sub-itemized suggestions are good, and revisions of the manuscript have been made in response, details follow below. But first I will address the topic of whether or not to keep Appendix B. My guiding principle here, is that the intention of including an appendix is to provide details for those who wish to go beyond a briefer description in the main text, and which is not available elsewhere. I have evaluated the question of keeping or dropping Appendix B from that perspective, while at the same time considering the reviewer's remarks and suggestions.

The reviewer correctly refers to his/her initial review ("Major revision item 4") in arguing that the original Sect. 2.3 was too long, and that an additional sub-section could be introduced in Sect. 3. However, in that review, there was a request for the text to be accompanied by a visual example (with no reference to an existing figure in that review). In commenting here on the present revision, the reviewer indicates that a reference could be made to Fig. 4 (thus not including a new figure, which I must admit to being certain was the original request).

My decision to include Appendix B in the first revision was also to give details in the form of definitions of edge curves and how modifications would affect the results of subsequent quantifications by validation metric values. A strong motivation for the level of detail was this statement in the reviewer's original report: *when adding these artificial "fixed" edges you automatically include in your verification some perfectly matched edges, aka trivial skill.* This is incorrect, and this motivated the introduction of Appendix B, and the level of details therein. However, the reviewer makes the same erroneous deduction above (*you automatically include in your verification some perfectly matched edges, aka trivial skill*; sub-item b here).

The only interpretation that involves perfectly matched edges, is the subsequent introduction of two metrics, where matching is introduced for both sets of ice displacement calculation. However, there is no trivial skill arising in this context either, so I can discard that interpretation.

I draw two conclusions from this: (1) Appendix B is needed, with the level of detail in the original revision retained (responding to sub-item c here). (2) Appendix B needs to be revised to reduce the risk that the readers, including the present reviewer, misunderstand how the metric is defined (sub-item b).

Let me here briefly explain the mistake. The calculations are based on computing distances from one edge curve (the "from edge") to another edge curve (the "to edge"). If coastal and open boundary edges were added to both of these edges, the reviewer's comment would be correct. However, additional edges are **only** added to the "to edge". The rationale is that the effect on validation results of the addition of edges should be kept to a minimum. And since the coastline and open boundaries are not included in the "from edge", no perfectly matched edges have been introduced. I here refer to Eq. (B3) and its explanation in the revised manuscript:
*Eq. (3) becomes*

$$\widetilde{d_n}^{\Delta t} = \min ||e_n^{(t_0+\Delta t)} - \widetilde{L}(t_0)||$$

*where $e_n^{(t_0+\Delta t)}$ is a grid cell on $L(t_0 + \Delta t)$, as before. Note that the set of grid cells $e_n^{(t_0+\Delta t)}$ is not affected, so the number of displacement distances considered in Eq. (5), $N(t_0 + \Delta t)$, is unchanged.*
The point here is that $e_n^{(t_0+\Delta t)}$ belongs to $L(t_0 + \Delta t)$, and not to $\widetilde{L}(t_0 + \Delta t)$. The devil is in the detail, here, the detail is in Appendix B, making a strong case for keeping the Appendix B. Hence my conclusion is to do just that. The new text in Appendix B that was mentioned above, follows this quote. **[P16L311-313]**

Above, I have responded to the reviewers comments under sub-items b and c. The reviewer gives a good argument for changing the structure slightly, and I have deleted sentences that refer to open boundaries and coastlines from Sect. 2.1 (sub-item f) and introduced a new section "3.2 Open boundaries and coasts" between the original Sect.s 3.1 and 3.2 (sub-item a). **[P8L179-P9L195]**

The text from lines 193-200 in the previous revision has been rephrased and inserted into the new Sect. 3.2, as suggested by the reviewer (sub-items d,f) **[P8L180-192]**. The end of the first paragraph in Sect. 3.3 (present version) has been rephrased following the reviewer's suggestion (sub-item f) **[P10L199-200]**. Statements regarding the modest impact of the addition of coastlines and open boundary grid nodes have been retained (sub-item g).

4. ***I won't recommend using the average*** *for describing an histogram (or a distribution) which is not gaussian (or even bimodal, as for the obs). I believe that* **the "mode" value (the one with the highest frequency),** *or even the median, would be better indicators. Please change lines 185-186 accordingly (talk about the mode and eliminate the discussion about the average). similarly, at lines 216-221: retain the first two sentences as they are (I like you essentially compare the accumulated frequencies for ranks below and above the 50%ile). The subsequent lines (218-221) are not entirely clear: do you use the methods of Wilks (2019)? if so, please put the reference up front and rephrase the whole text, so that it becomes more clear. Similarly, at lines 226-229 you perform an identical analysis, but you are missing the conclusions (one sentence, stating that these findings show that the model well depicts the position of the max displacement).*

I agree that the average values are not the best choice here. As the set of values depend on the resolution and grid orientation, I think that the mode value is not the best option. Also, a category mode can already be inferred from Fig. 5. Hence, I choose to specify median values in the text. **[P10L209-210]**

None of the methods of Wilks (2019) were used in the previous versions of the manuscript. As a response to this reviewer comment, I have now performed the $\chi^2$ test, and the results are given in the present revision. This test confirms the conclusion from the previous analysis, as the null-hypothesis of a flat histogram is firmly rejected. **[P11L231-233]**

**Minor Revisions**

5. *Some suggested re-phrasing in the Introduction:*

   a *line 18 and 19, rephrase "in their examination" (e.g. "to analyse the predictability of sea-ice edge")*
   Rewritten following the reviewer's suggestion. **[P1L18]**

   b *line 37-38, write "The method is described in Section 2 with an idealized case study."*
   Rewritten as suggested by the reviewer. **[P2L37-38]**

   c *line 38, replace "in an examination of displacements" with "to analyse displacements ... "*
   Rewritten as suggested by the reviewer. **[P2L38]**

   d *line 39: write "Technical details ... "*
   Rewritten as suggested by the reviewer. **[P2L39]**

6. *rephrase lines 47-48.*

   The sentence in these lines is basically a repetition of the paragraph's first sentence, and are not needed. So rather than rephrasing, I have decided to delete the sentence in question.

7. *line 60: replace "line" with "curve" (possibly everywhere, since ice edge is rarely straight).*

   I have chosen to follow this advice where relevant. (I have kept "line" where it refers to a straight line. I still use the word "coastline" to indicate the separation between land and sea, even when this is not a straight line.) **[througout]**

8. *Suggested rephrasing for lines 67-74: line 67, end this sentence after "respectively"; then place Equation (4) followed by the text at line 70 as "where we denote the sea ice concentration ... (t0)." Then you start a new paragraph introducing Figure 1: please expand a bit with respect to the text in line 68-69, and then concatenate with the text at lines 73-74. Example, start with Figure 1 shows an idealized example where a modeled and an observed sea ice edge are displaced. The length of the dashed lines correspond to ... . We then introduce the maximum expansion displacement as ...").*

   I completely agree with all of these comments. The suggestions have been adopted, improving the structure and readability of the relevant paragraphs. **[P4L67-72]**

9. *rephrase lines 91-93.*

   I agree that these lines should be rephrased. But in doing so, I came to the conclusion that the best option was to rephrase the entire paragraph (lines 89-93). **[P4L91-94]**

10. *In light of major comment 3, eliminate lines 94-97.*

    Lines 94-97 have been removed.

11. *I suggest eliminating also lines 98-99, and move this comment in the conclusions.*

    I agree. In fact, this comment was already included in the conclusions in the previous revision (lines 241-243). **[P13L260-261]**

12. *lines 108-111 repeat the same concepts stated in the previous paragraph (lines 104-107). Eliminate one of the two paragraphs, but (in light of Major comment 2) I would like you to expand and describe the result of Figures 2 and 3 more in detail.*

    The discussion of the results displayed in Figures 2 and 3 have been rewritten from lines 104-111 in the previous revision, and some more details are given in the present revision. I have also added the median values in the caption to Fig. 2, following the reviewer's advise in Major revision item 4 above. **[P5L101-109, Fig. 2]**

13. *please rephrase lines 145-146, e.g. "we randomly subsample a fixed number of intervals from Eq (11), so that the number bins is equal across different cases and results can be aggregated".*

    Rewritten as suggested by the reviewer. **[P7L142-144]**

14. *line 149, add "which reflects a forecast poor positioning of the maximum displacement"*

    Rewritten following the reviewer's suggestion. **[P7L147-148]**

15. *eliminate lines 191-192 (it seems out of context here). Maybe this sentence is more suited in the data description, section 3.1?*

    I agree. This topic is now addressed in Sect. 3.1, and the paragraph in the original revision where the ice chart data are described have been split into three smaller paragraphs, intending to make the information more easy to absorb for the readers. **[P8L167-169]**

16. *Rephrase lines 202-203, e.g. "We consider a fix number of 10 bins for the present investigation, Hence ... nine values are randomly selected from the displacements in Eq. (11)".*

    Rewritten as suggested by the reviewer. **[P10L213-215]**

17. *Lines 214-215 are not clear, rephrase (or eliminate) them.*

    The lines have been eliminated from the main body of the text, as suggested by the reviewer. A rephrased version, which should be clearer, has been included in the caption of Fig. 7. **[Fig. 7]**

18. *Line 216: rephrase "in ranks 5-9 than ranks 0-4".*

    Corrected according to the reviewer's comment. **[P11L226]**

19. *The first paragraph of the conclusions is weak, and can be improved.*

    This paragraph has been rephrased. Note that this paragraph was also modified in response to a suggestion from Referee Report 1. **[P13L249-251]**

20. *In the text at lines 248-252 state explicitly that ocean open boundaries and coastal lines become part of the ice edge (rather than stating "a modification to the algorithm was introduced").*

    This text has been modified, taking the response to Major revision item 3 into account. **[P13L270-272]**

21. *Figure 7: why when considering the whole domain there is a peak in the 4-5 ranks, whereas when considering the two separate domains this disappear? Are there still mis-matched ice edges?*

    The differences between the distributions in panels (a) and (b) are due to a combination of effects: a smaller rank size in the subdomains, introduction of an additional displacement maximum for each analysis date, and the introduction of a new open boundary (the separation line between the two sub-regions). The reviewer's observation of "missing peaks" illustrates a situation that can arise due to these effects. This well-spotted contrast is mentioned in the present revision, along with a description of the effects of changing from the full domain view to subdomains. Once more, the reviewer's constructive comments have improved the manuscript. **[P12L238-244]**

    I'm not sure what the reviewer refers to with the final question concerning mismatching ice edges. In this context, I refer to the statement at the start of Sect. 3.3 (present revision), which explicitly states that the analysis in this section is performed according to the expansion in Sect. 3.2/Appendix B.

---

## Author Response (AR3)

**Author's response for the handling Editor**

Dear Dr. Kaleschke,
I am very grateful for the outcome of the review process. Please find below a few minor comments regarding the present revision.

*I have a final request:* This reviewer has gone to great lengths to raise the quality of my manuscript, through multiple long referee reports with detailed suggestions for improvements. As a consequence, the present revision has changed substantially from my initial submission, all to the better. I have been a co-author to several papers when contributing less than this reviewer. So I'm asking you to approach the reviewer and offer him/her to become a co-author on this paper. (Fees will be paid in full by my institution.)

Should this invitation be accepted, the information in the manuscript where authors and their contributions are mentioned will obviously need to be modified.
Best regards,
Arne Melsom

**Author's comments**

I have adjusted the manuscript in all places where recommended. Note that in some places I have changed the suggestions for consistency with the existing text, e.g. writing *analyzed* (not *analysed* [L98]), *mismatch* (not *mis-match* [Sect.3.2; L267-268]), *validation* (not *verification* [Sect.3.2]). I also have made some additional minor modifications to some suggesions, these can all be easily identified in the "diff" file. But I must underscore that generally, introducing the suggested changes has lead to an improved manuscript.

There is one place where I must admit that I'm very much in doubt. This is the suggestion
**line 246: start the sentence with "Whereas, the ranges..."**
This is a case of one statement followed by another. In this case, the second statement is a modification of the first statement. It is not the opposite of the first statement, and my English skills are insufficient to decide whether or not *Whereas* is correct in this context. I decided to follow the reviewer's advice in the present revision.